# Structural basis for DNA break sensing by human MRE11-RAD50-NBS1 and its regulation by telomeric factor TRF2

Yilan Fan [1,3], Filiz Kuybu [1,3], Hengjun Cui [1,3], Katja Lammens [1], Jia-Xuan Chen[2], Michael Kugler [1], Christophe Jung [1] & Karl-Peter Hopfner [1] ✉

The MRE11-RAD50-NBS1 (MRN) complex is a central, multifunctional factor in the detection, signaling and nucleolytic processing of DNA double-strand breaks (DSBs). To clarify how human MRN binds generic and telomeric DNA ends and can separate DNA end sensing from nuclease activities, we determined cryo-electron microscopy (cryo-EM) structures of human MRN bound to DNA and to DNA and the telomere protection factor TRF2. MRN senses DSBs through a tight clamp-like sensing state with closed coiled-coil domains, but auto-inhibited MRE11 nuclease. NBS1 wraps around the MRE11 dimer, with NBS1's ATM recruitment motif sequestered by binding to the regulatory RAD50 S site, necessitating a switch in the NBS1 C helix for ATM activation. At telomeric DNA, TRF2 blocks the second S site via the iDDR motif to prevent nuclease and ATM activation. Our results provide a structural framework for DNA sensing via a gating mechanism and separation of sensing, signaling and processing activities of mammalian MRN.

DNA double-strand breaks (DSBs) spontaneously arise from ionizing radiation, genotoxic chemicals, abortive topoisomerases or replication fork collapse, but are also programmed intermediates in meiotic recombination[1,2] and immunoglobulin gene rearrangements[3–5]. Failure to properly repair DSBs can cause chromosomal aberrations, death or cancer[6,7]. Cells respond to DSBs through a cell cycle-regulated DNA damage response (DDR), primarily mediated by the activation of ATM, ATR, and DNA-PK kinases. This response leads to complex chromatin modifications[8,9] and promotes either classic non-homologous end joining (c-NEHJ) or long-range resection followed by homologous recombination (HR)[10–16]. When both NHEJ and HR fail, cells can utilize error prone alternative end joining (alt-EJ) pathways[17].

A central sensor of DSBs is the MRE11–RAD50–NBS1 (MRN) complex, known as MRX (Mre11-Rad50-Xrs2) in *S. cerevisiae*. MRN and its prokaryotic MR homologs are evolutionarily conserved ATP-dependent complexes with nuclease activity[18–24]. The core MR complex contains two copies of both the ATPase RAD50 and the endo-/exonuclease MRE11, assembled in an elongated structure. This structure consists of a "head" module which includes MRE11 and the ATP binding cassette type nucleotide binding domains (NBDs) of RAD50, and a "tail" module formed by the coiled-coil domains (CCDs) of RAD50[25–29]. NBS1[18] is only found in eukaryotes, where it recruits ATM[30,31] and the MRN nuclease cofactor CtIP[32–34], among other factors. The CCDs of RAD50 are apically joined generating a ring-like protein structure that serves as an ATP-dependent gate for recognizing single DNA ends[35–39]. In *Escherichia coli* (*Ec*) MR (known also as SbcD), ATP binding opens the CCDs to load MR onto linear DNA, while ATP hydrolysis closes the CCDs to a rod-like state that repositions SbcD[MRE11] from an autoinhibited "resting" to a "cutting" state[37,40].

Following the detection of DSBs, MR/MRN nicks the 5' strand in the vicinity of the break[2,41–48], possibly followed by back-resection through its 3'→5' exonuclease activity[24,49], as well as cleavage of the exposed 3' strand[45,50]. The nuclease activity removes covalent DNA-protein adducts[1,48], opens and degrades hairpin structures, can

[1]Gene Center, Department of Biochemistry, Ludwig-Maximilians-Universität München, Feodor Lynen Straße 25, 81377 Munich, Germany. [2]Proteomics Core Facility, Institute of Molecular Biology, Mainz, Germany. [3]These authors contributed equally: Yilan Fan, Filiz Kuybu, Hengjun Cui. ✉e-mail: karlpeter.hopfner@lmu.de

generate ssDNA in alt-EJ pathways[51,52] and activate the long-range resection machinery in HR[16].

MRN/X has highly regulated, multifunctional roles in the DDR that besides sensing and nuclease activities also includes DNA tethering and ATM/Tel1 activation. Furthermore, MRN/X's nuclease is highly regulated by trans-interacting factors that interact with a regulatory surface patch on RAD50, denoted "S site". Structure prediction and functional studies suggest that the co-activator CtIP/Sae2 binds the S site to promote 5′ incision, while at telomeres, TRF2 (human) or Rif2 (yeast) bind the S site to prevent MRN-dependent activation of ATM and counteract CtIP/Sae2[53–61].

To provide a structural framework for sensing of DNA ends by human MRN and its regulation, we determined cryo-EM structures of MR and MRN bound to DNA, as well as bound to DNA and TRF2. Our work identifies a "sensing state" that in this form is not observed in prokaryotic MR and can explain decoupling of MRN's structural and nucleolytic functions at DSBs. Our structure shows that NBS1's conserved C-terminal region (referred to as C helix in our structure), which carries the ATM recruitment motif, is sequestered at the RAD50 S site, with amino acids that were previously found to bind to ATM. This observation suggests that an NBS1 C helix switch is needed to form the MRN-ATM complex. We also provide a structural basis for the telomeric TRF2-MRN complex, showing that both RAD50 S sites are blocked through NBS1 on one side and the TRF2 iDDR motif on the other side of the dimer.

## Results

### Structural basis for MRN's interaction with DNA

Subunits of the MRN complex (Fig. 1A) were co-expressed in human or insect cells (Supplementary Fig. 1A, see Methods). The purified MRN complex showed the characteristic DNA-stimulated ATPase activity (Fig. 1B) and interaction architecture, determined by mass photometry and chemical cross-linking coupled with mass spectrometry (CX-MS) (Supplementary Fig. 1B–D)[62]. To obtain DNA-bound structures of MRN, we screened several different types of DNA and nucleotides. The highest resolution structures were obtained by incubating MR or M^H129N^RN (nuclease dead mutant, hereafter referred to as 'MRN') with 50 bp DNA and ATP at 35 °C for 30 min, followed by addition of BeF$_x$ (producing the ATP mimic ADP•BeF$_x$ after ATP hydrolysis). Extensive processing resulted in 3D reconstructions with overall resolutions of 3.2 Å (MR) and 3.1 Å (MRN) (see Methods and Supplementary Figs. 2–4). The maps resolve MR's and MRN's catalytic head bound to DNA and parts of the CCD module, with ADP•BeF$_x$•Mg$^{2+}$ bound at both active sites (Fig. 1C-F). We do not observe density for MRE11's C-terminal extension (510-708), the N-terminal region of NBS1 (1-650), both of which are predicted to be unstructured or mobile, and the remainder of the RAD50 CCDs (~245-~1070). Atomic models were generated by rigid body fitting AlphaFold3 predictions[63], manual fitting and automated refinement.

Human MR/MRN comprises two tightly engaged RAD50^NBD^s that bind into the concave side of the MRE11 dimer (Fig. 1C, E). Furthermore, MRE11 interacts with RAD50's CCDs via a helix-loop-helix (HLH) domain, which is expanded to four helices in the human structure (four-helix bundle). The nuclease active site of MRE11 harbors density consistent with two Mn$^{2+}$ ions (present in the buffer, Supplementary Fig. 4A) but is shielded from any DNA access by the proximity of RAD50^NBD^s.

DNA interacts with both RAD50^NBD^s in a two-fold symmetric manner, traversing the RAD50 dimer between the CCDs with a footprint of ~20 bases (Fig. 1C, E, Fig. 2A, D–F), consistent with ATP dependent DNA binding (Fig. 2C). The 3′ strand, as viewed from a DSB, binds to the RAD50^NBD^ N-lobe along the top β-strand of the S site and to the subsequent α-helix, while the 5′ strand binds the N-lobe at a loop following the P-loop helix (Walker A motif) (Fig. 2D). Further contacts are observed at the RAD50^NBD^ C-lobe, where the 3′ strand binds to a

helix connecting the Q-motif and CCD and a hairpin loop inserts R1179^RAD50^ into the minor groove (Fig. 2E, Supplementary Fig. 4B). MRE11 does not substantially contribute to DNA binding in this state, with only a single residue (K467^MRE11^) at the HLH element in sufficient proximity to the 3′ strand. DNA interactions are virtually identical in MR and in MRN complexes. However, the highly positively charged C-terminus of NBS1 (K$_{751}$KRRR$_{755}$) binds the DNA emerging from RAD50 and induces additional DNA contacts on one side of MRN (Fig. 2F).

Insertion of the RAD50^NBD^ hairpin loop (1168-1183) into the minor groove goes hand in hand with an inward tilting of both CCDs, which meet ~20 Å above the DNA (Fig. 2A, Supplementary Fig. 4E). Together, RAD50^NBD^s and CCDs form a tight clamp around a single DNA duplex, supported also by CX-MS (Supplementary Fig. 1C, D). Together with AFM studies on human MR/MRN with DNA[38] and crystallographic analysis of the zinc-hook region[36], both showing rod-like conformations, the structure argues that CCDs may adopt a gate for a single loose DNA end like prokaryotic MR^SbcCD[37,40]^. Here, restricting DNA passage through the CCD rod to a single DNA duplex ensures that the stable clamp can only form after loading onto linear DNA, but not internal or circular DNA. In latter case, at least two dsDNA segments would traverse the CCDs, preventing full rod formation through steric hindrance.

### A "sensing state" decouples "resting" and "cutting" conformations

We observe density for ADP•Mg$^{2+}$, with the better resolved structures also showing density for BeF$_x$ (Fig. 1D). ADP•Mg$^{2+}$•BeF$_x$ moieties are bound to Walker A and B motifs of one RAD50^NBD^, and to the signature motif helix on the other RAD50^NBD^ (Supplementary Fig. 5A). H1269^RAD50^ is positioned to act as γ-phosphate sensor.

In several datasets, we observe 2D classes with open CCDs, particularly in the presence of ATPγS, more rarely in the presence of ATP +BeF$_x$ (Supplementary Fig. 6A–G). In latter case, the MRE11 dimer is also sometimes tilted with respect to RAD50^NBD^s (Supplementary Fig. 6C), indicating some mobility between RAD50 and MRE11 dimers. A 3D reconstruction of MRN in the presence of DNA and ATPγS resulted in an anisotropic map (Supplementary Fig. 7). While the anisotropy prevented detailed modeling, we could convincingly rigid-body fit an AlphaFold3 model of MRN-ATP, where the CCDs have been truncated in silico (Fig. 2B, Supplementary Fig. 8). We did not observe DNA in this state despite its presence in the sample. Since MRN shows similar binding affinity towards DNA in the presence of ATP and ATPγS (Fig. 2C), perhaps a faster on – off kinetics between DNA and MRN in the resting state with open CCDs may lead to DNA dissociation during sample blotting and plunge freezing. In any case, ATPγS binding to MRN can widen the CCDs like the resting state of SbcCD[37]. DNA binding and possibly ATP hydrolysis prompt closure of the CCDs to form the clamp. However, in contrast to EcMR, closing of the CCDs does not yet reposition the MRE11 nuclease from the auto-inhibited state to the cutting state. We therefore refer to this new clamp state observed in human MR and MRN as the "sensing" state.

### NBS1 tethers MRE11 to the RAD50 S site via the ATM recruitment motif

Our structure resolves a single NBS1 polypeptide (651-754), which wraps around the MRE11 dimer with the highly conserved K$_{683}$NFKKFKK$_{690}$ motif situated across the two-fold symmetry axis (Fig. 1F, Fig. 3). NBS1 does not induce significant changes into the DNA-bound MR structure, compared to MR alone (Supplementary Fig. 6H).

Intriguingly, the very C-terminal region of NBS1 (718-745) forms a continuous helix ("C helix") that bridges one MRE11 catalytic domain to one RAD50^NBD^ in the sensing state (Fig. 3A). The first part of the helix binds MRE11 via a hydrophobic interface between L719^NBS1^, L723^NBS1^, W722^NBS1^ and F233^MRE11^ (Fig. 3C). The second part of the C helix is bound to the N-lobe β-sheet at the RAD50 S site (Fig. 3D). Here F744^NBS1^ binds

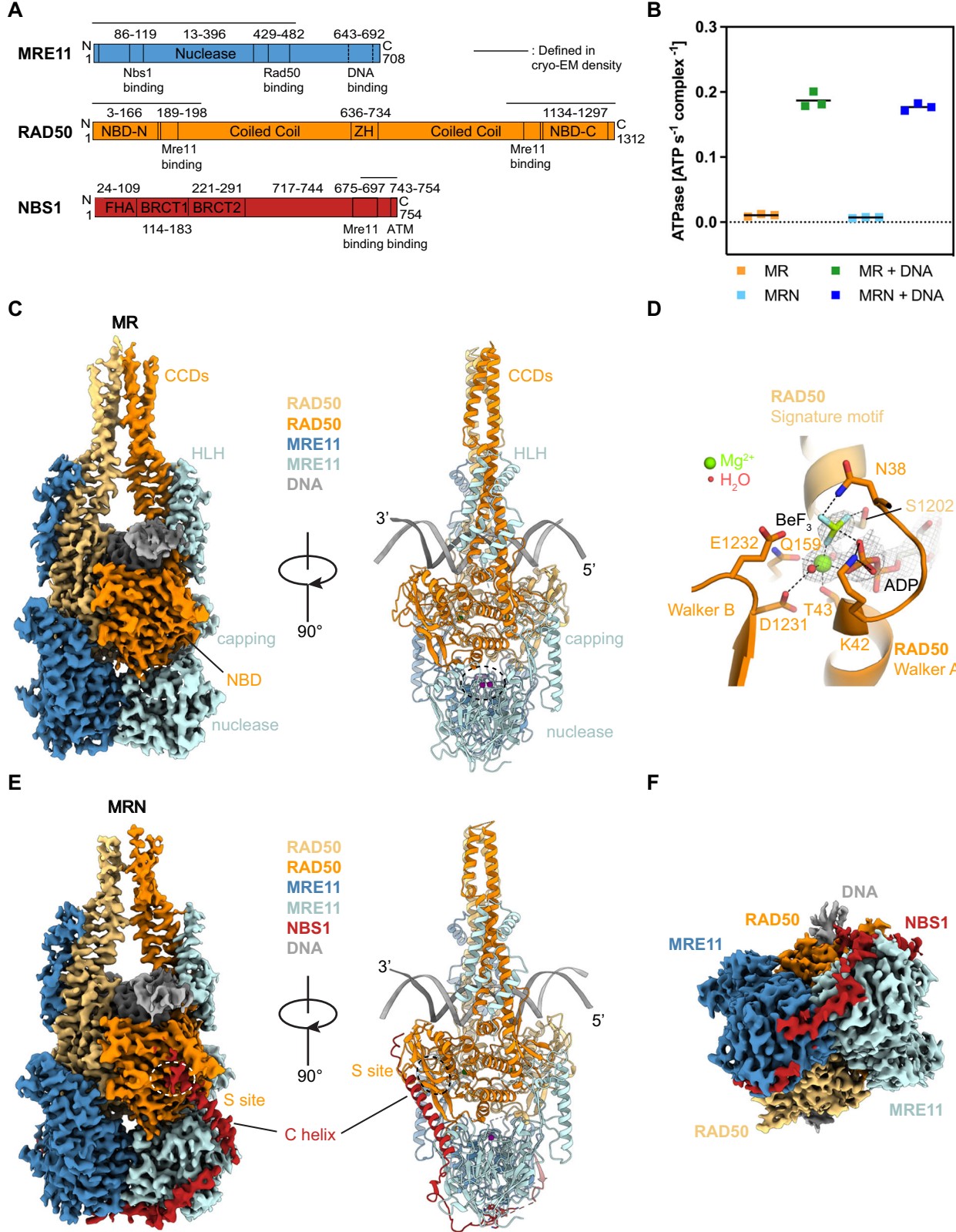

into an S site pocket lined by K6[RAD50], S8[RAD50], I24[RAD50], R83[RAD50] and Q85[RAD50]. Further notable interactions are electrostatic interactions between K6[RAD50] and D741[NBS1], and between R83[RAD50] with the dipole charge of the NBS1's C helix.

During 3D classification, we identified two separable conformational states in MR and MRN, one with the MRE11 dimer symmetrically attached to both RAD50[NBD]s, and one where the MRE11 catalytic

domain dimer detaches as a rigid body from one RAD50[NBD] (Supplementary Figs. 2, 3; Supplementary Fig. 6C, D). In the case of MRN, the detachment is only observed at the RAD50[NBD] not bound to NBS1. These observations suggest a stabilizing effect of the NBS1 C helix-S site interaction on MRN catalytic head dynamics.

Since the C helix contains an important ATM recruitment motif[30,31,64] we compared S site binding of the C helix to binding of the

**Fig. 1 | Overall structure of the human MR(N) complex bound to DNA. A** Domain arrangement of human MRN subunits. Domain boundaries are indicated with numbers and vertical black lines. ZH, zinc hook; FHA, forkhead-associated domain; BRCT, breast cancer C-terminal domain; NBS1 C helix, NBS1 C-terminal helix. **B** ATPase activity of MR(N) complex stimulated by DNA binding. Data are presented as mean and individual data points from three technical replicates. **C** Overview of the MR-DNA complex structure. The high resolution cryo-EM map is displayed (left panel). The MRE11 nuclease active site (buried) is highlighted with a white/black dashed circle. **D** Ribbon representation of the nucleotide-binding site of Rad50

highlighting the presence of BeF$_3$ bound together with ADP. The density of ADP, BeF$_3$, Mg$^{2+}$ and water molecules from the structure of MR-TRF2$^{iDDR-Myb}$-DNA complex (which has the best resolution) was contoured at 9 σ level. Black dashed lines represent electrostatic interactions or hydrogen bonds. **E** Overview of the MRN-DNA complex structure. High resolution cryo-EM density was displayed (left panel). The RAD50 S site is highlighted in white/black dashed circle. **F** Top view of the MRN-DNA complex structure illustrated in **E** (left panel) obtained by a 90-degree rotation around the x axis. Source data for **B** are provided as a Source Data file.

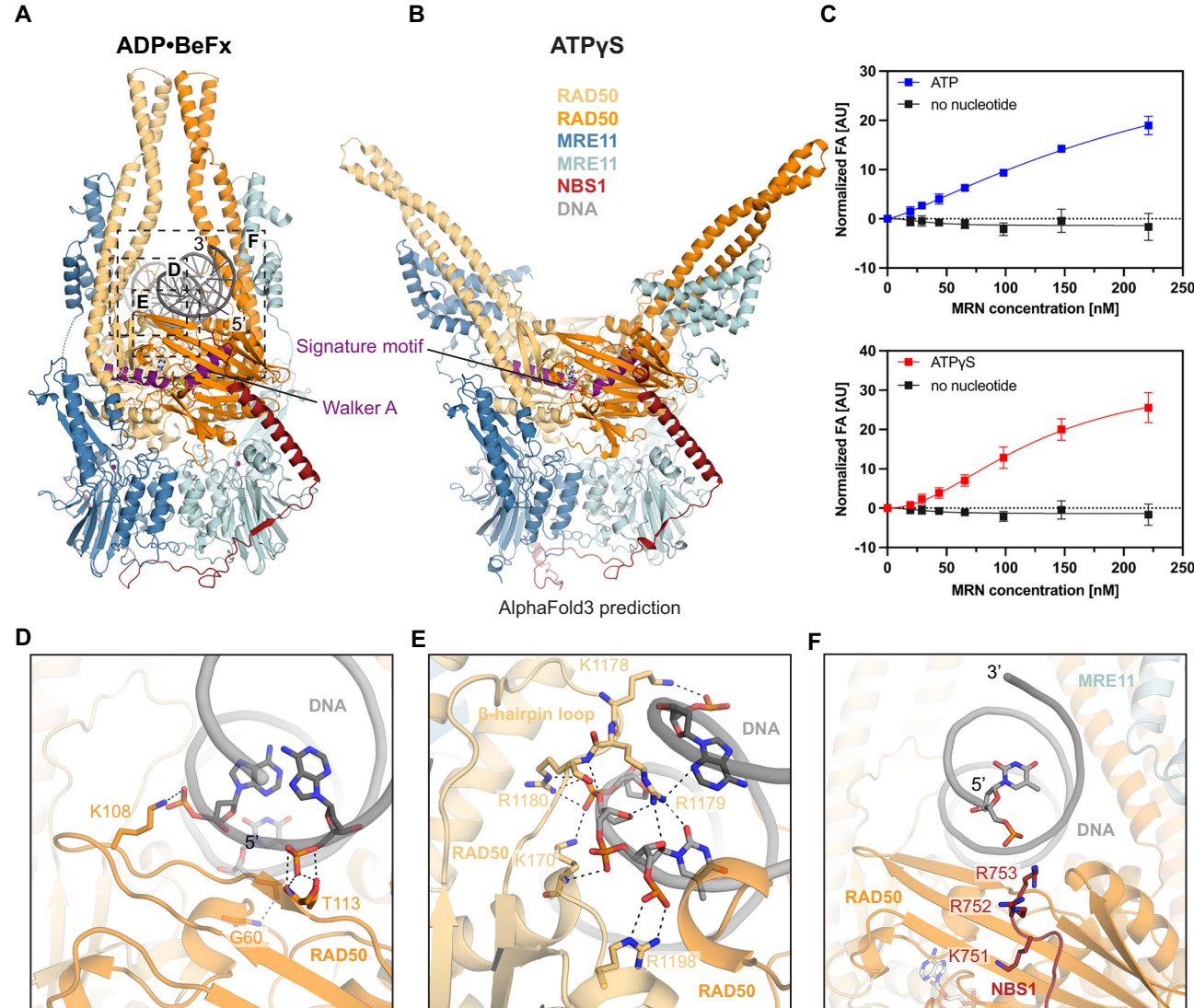

**Fig. 2 | Structural basis of DNA binding by human MRN complex. A**, **B** Side by side comparison of the MRN-DNA-ADP•BeF$_x$ (sensing state) and MRN-ATPγS (resting state) complex structures. The α-helices containing either the Walker A or signature motifs in both structures were colored in purple for better visualization. **C** Fluorescence anisotropy (FA)-based assay documenting the effect of ATP and ATPγS on MRN binding towards 64 bp Myb DNA. Data are presented as mean values

+/− SD from three technical replicates. **D**–**F** Detailed molecular contacts between DNA and the MRN complex. For simplicity, only the nucleotides of the DNA template involved in the interactions were displayed as sticks. Black dashed lines represent electrostatic interactions or hydrogen bonds. Source data for **C** are provided as a Source Data file.

NBS1 C-terminal peptide to the ATM dimer (PDB accession code: 7SID)[65] (Fig. 3D, E). Both interactions are mediated by a highly conserved DLFRY sequence in NBS1 (Fig. 3B). F744 at the center of the motif provides a hydrophobic anchor in both MRN and ATM/NBS1 structures, binding to pockets in either the RAD50 S site or in the regulatory N-terminal HEAT repeat region of ATM. Evidently, RAD50 in the sensing state and ATM would need to compete for the C helix and ATM recruitment by MRN necessitates detachment of the C helix of

NBS1 from the S site. Altogether, this suggests a conformational change in MRN and detachment of NBS1 from RAD50 in the DNA sensing state to a signaling state in complex with ATM.

## Structures of the TRF2-MR and TRF2-MRN complex bound to DNA

To provide a structural basis for inhibition of ATM signaling and DNA cleavage at telomeric DNA, we determined cryo-EM structures

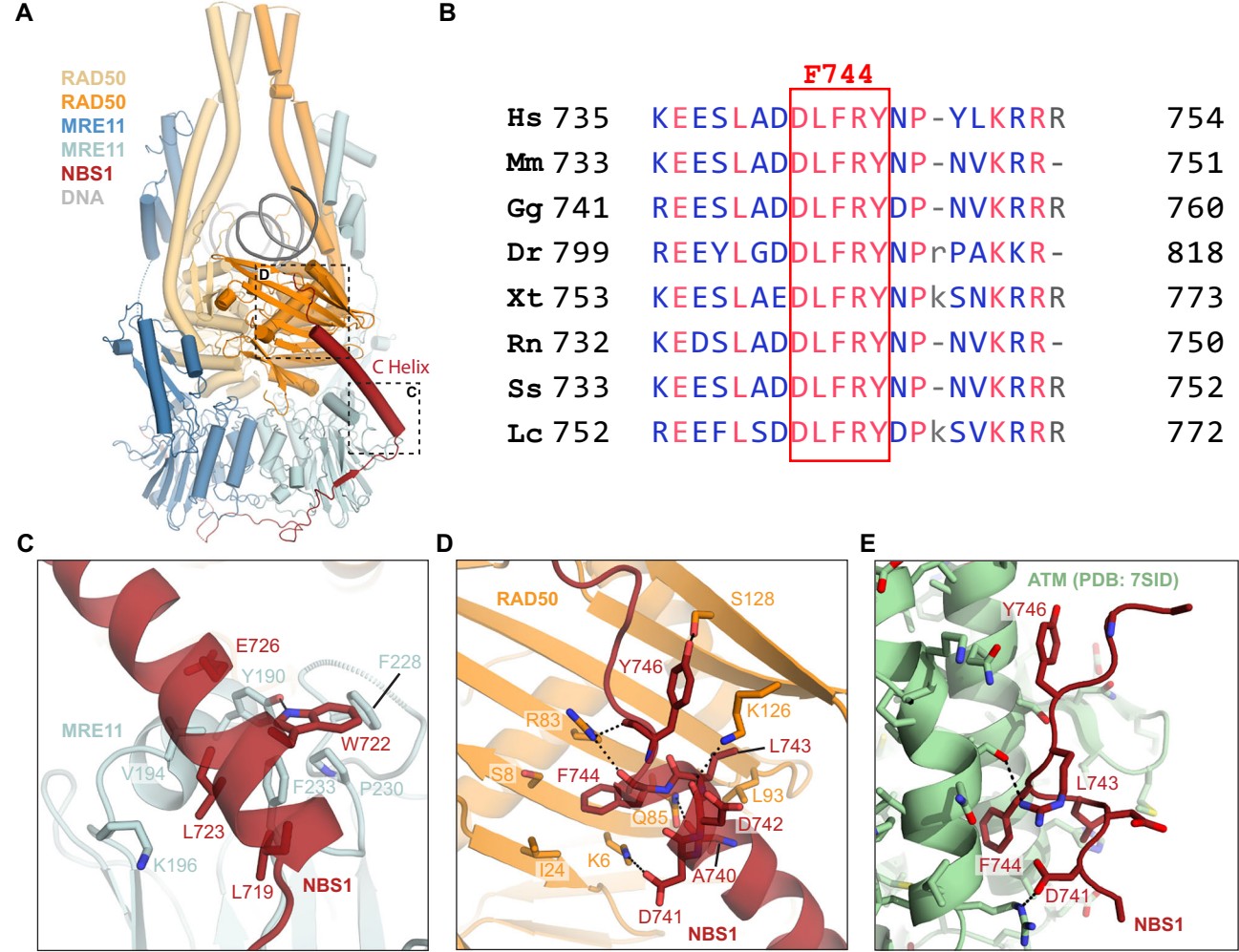

**Fig. 3 | NBS1 interactions with the MR complex mediated by the C helix.**
**A** Overview of the MRN-DNA complex structure in ribbon model. **B** NBS1 sequence alignment (*Hs*: *Homo sapiens*, *Mm*: *Mus musculus*, *Gg*: *Gallus gallus*, *Dr*: *Danio rerio*, *Xt*: *Xenopus tropicalis*, *Rn*: *Rattus norvegicus*, *Ss*: *Sus scrofa*, *Lc*: *Latimeria chalumnae*) highlighting the conservation of F744 (numbered according to *Hs*NBS1). **C**, **D** Close-up views of the interactions formed between NBS1 C helix and MRE11 or RAD50, respectively. Only interacting residues were displayed as sticks here for simplicity, and the black dashed lines represent electrostatic interactions or hydrogen bonds. **E** Associations of NBS1 C helix with ATM (PDB: 7SID) illustrated in a similar view to **D** for the purpose of side-by-side comparison.

of MR and MRN bound to the shelterin[66,67] component TRF2 and DNA containing TTAGGG motifs (Fig. 4A–D, Supplementary Fig. 9-11). TRF2 interacts with the S site via the iDDR motif based on structure prediction and mutagenesis, but a comprehensive architectural understanding including the conformational state of MR or MRN remains open[53,61]. TRF2 contains a telomeric repeat factor homology (TRFH) dimerization domain, an iDDR motif, and Myb domain (490-542) that binds double-stranded telomeric TTAGGG repeats. Adding the TRF2[iDDR] (438-485) to our cryo-EM sample preparation procedure led only to free MRN-DNA and MR-DNA structures without additional density for TRF2[iDDR]. We also did not observe an interaction of TRF2[iDDR] with MRN in gel filtration, although more sensitive yeast two-hybrid assays can detect RAD50 TRF2 interactions[61]. We reasoned that by including a Myb domain, we can stabilize the interaction through avidity effects, essentially recapitulating what happens at telomeric DNA. In such a scenario, tight binding of TRF2 to the telomeric DNA sequence through the Myb domain would increase the local concentration of iDDR in the vicinity of MRN that also binds to telomeric DNA, essentially populating iDDR-MRN interactions despite low $K_{Ds}$. Using a TRF2 construct harboring both the iDDR and Myb domains (TRF2[iDDR-Myb]), as well as full-length TRF2, we could indeed determine high-

resolution structures of MR-TRF2[iDDR-Myb]-DNA, MRN-TRF2[iDDR-Myb]-DNA and MRN-TRF2-DNA (Supplementary Figs. 9–11).

The reconstructions show well-resolved density for TRF2 459-473, essentially comprising the iDDR region (Fig. 4A, B). Myb and TRFH domains are not resolved, indicating a flexible and structurally independent arrangement that averages out in 3D reconstructions. The core of the iDDR S site interaction is a short helical element that forms a hydrophobic-aromatic anchor through W465[TRF2], L471[TRF2], and F472[TRF2] with a small hydrophobic patch on RAD50 (L93, A95, V117). An acidic cluster ($E_{467}EDE_{470}$[TRF2]) contributes to the core interaction by forming electrostatic interactions with K6[RAD50], K22[RAD50], R83[RAD50], and K126[RAD50]. The region C-terminal to this core interaction points toward the DNA section exiting the MR/MRN head, consistent with the Myb domain binding to DNA in the wider vicinity of MRN. We observe density up to Q475[TRF2]. The subsequent 15 amino acid linker and Myb domain C-terminal to the RAD50 binding region, are not resolved.

In the case of MR, both RAD50[NBD]s are occupied by iDDR, while in the case of MRN, density for iDDR was only observed at one RAD50 S site, whereby the other one is still bound by the NBS1 C helix (Fig. 4D). This results from steric hindrance rather than allosteric regulation, as NBS1 physically blocks binding of a second TRF2 iDDR molecule. While the binding sites of NBS1 and TRF2 strongly overlap (Fig. 4C), with

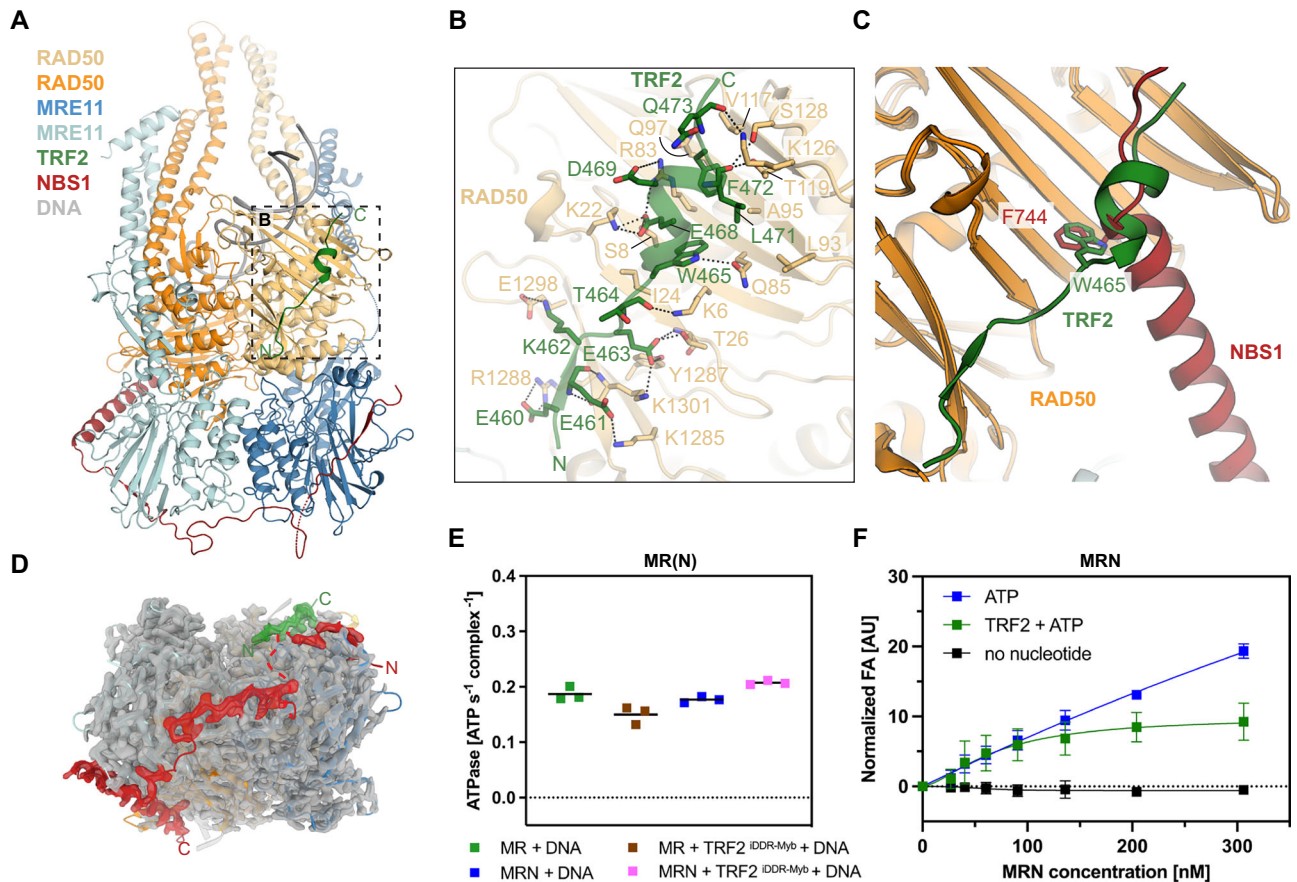

**Fig. 4 | Structural basis of TRF2^iDDR-Myb binding to the human MRN complex.**
**A** Overview of the MRN-TRF2^iDDR-Myb-DNA complex structure. **B** Zoomed-in view of the molecular interactions between TRF2^iDDR and RAD50. The residues involved in the associations were illustrated as sticks, black dashed lines represent electrostatic interactions or hydrogen bonds. **C** Comparison of binding sites to RAD50 between NBS1 C helix and TRF2 centering the evolutionary conserved hydrophobic residue F744 (*Hs*NBS1) or W465 (*Hs*TRF2). **D** Top view of the MRN-TRF2^iDDR-Myb-DNA complex structure emphasizing the asymmetric binding property of NBS1 and TRF2.

The high resolution cryo-EM density is displayed in grey except for that of the resolved parts of NBS1 and TRF2, which were colored in red and green, respectively. **E** Comparison of ATPase activities by MR/MRN complex in the presence or absence of TRF2^iDDR-Myb. Data are presented as mean and individual data points from three technical replicates. **F** Fluorescence anisotropy (FA)-based assay documenting the effects of ATP, as well as ATP and TRF2 on MRN binding towards 64 bp Myb DNA template. Data are presented as mean values +/− SD from three technical replicates. Source data for **E** and **F** are provided as a Source Data file.

hydrophobic anchor residues (F744^MRE11 or W465^TRF2) occupying the same S site pocket, "MR" and "MRN" parts of the TRF2 complexes are practically identical to the corresponding structures obtained in the absence of the TRF2/TRF2^iDDR-Myb. The similarity includes also nucleotide state and active site geometry (Supplementary Fig. 12C). Consistently, we neither observe a stimulation nor a reduction of MRN's ATPase by TRF2^iDDR-Myb (Fig. 4E). Along the same lines, in TRF2/TRF2^iDDR-Myb complexes, we also observe the mobility of MRE11 dimer in relation to Rad50, which detaches from the TRF2-bound but not the NBS1-bound RAD50^NBD protomer in the complex (Supplementary Fig. 6E-G, I).

The "rigid-body"-like recognition of the sensing state by TRF2, along with its lack of impact on MRN ATP hydrolysis, suggests that TRF2 does not directly provoke MRN release from DNA. To test this, we performed DNA binding assays in the presence of ATP. TRF2 bound to DNA with a $K_D$ of 121 nM (Supplementary Fig. 12A). Titrating MRN in the presence or absence of a saturating concentration of TRF2, which is prebound to DNA to mimic the situation at telomeres, resulted in MRN binding to DNA in both cases. This suggests that even saturating concentrations of TRF2 do not block MRN DNA binding. We could not reach saturating MRN binding in the absence of TRF2 due to a general limit to concentrate MRN. Possible causes are multiple binding events or protein clustering, as has been observed for MRX[68]. However, we

obtain a properly saturated binding curve in the presence of TRF2 (Fig. 4F) suggesting that TRF2 promotes more defined binding of MRN.

Taken together, TRF2 binds MRN in a 1:1 complex (2 MRE11, 2 RAD50, 1 NBS1, 1 TRF2) and prevents ATM and CtIP associated activities by saturating and blocking the free RAD50 S site, without impacting the sensing state itself.

## Discussion

Here we provide cryo-EM structures of the human MRE11-RAD50-NBS1 (MRN) DSB sensor and repair factor bound to DNA and TRF2, altogether providing a structural framework for DNA end sensing by MR/MRN at generic DNA double-strand breaks or at dysfunctional telomeres, which have lost full protection through shelterin and T-loop formation as well as expose a loose DNA end. The resting structure with open CCDs and sensing structure with closed CCDs are generally consistent with a "topological" sensing of DSBs as proposed for the *Ec*MR^SbcCD homolog. Here, DNA topology differentiates loose DNA ends from protected DNA ends such as telomeric T-loops, as well as undamaged chromosomal DNA stretches. It remains to be shown whether the Rad50 CCDs form a rod all the way to the zinc-hook in the sensing state. However, full rods are observed upon DNA binding in atomic force microscopy[38], and the zinc-hook regions is also consistent with a rod conformation[36]. Together, these studies argue that DSBs are

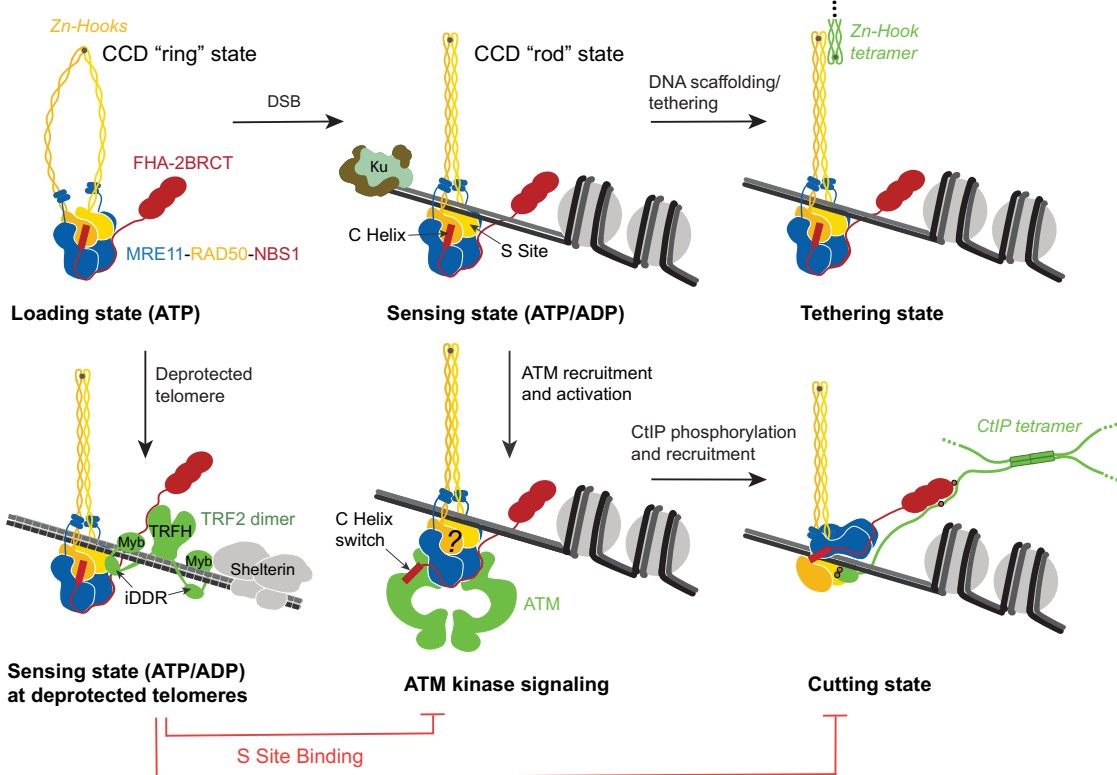

**Fig. 5 | Proposed working model of the MRN complex initiated by DNA double-strand breaks.** MRN senses DSBs at generic or at de-protected telomeres by topologically detecting DSBs through ATP-dependent transitions of the CCDs from a ring to a rod-like conformation, followed by loading onto the linear DNA. Upon DSB recognition, MRN recruits and activates ATM kinase, inducing conformational changes in the complex. CtIP is then phosphorylated and recruited to MRN at the DSB site, promoting the transition to the cutting state for DNA repair. At dysfunctional telomeres, the TRF2-bound sensing state of MRN represents a nuclease-inhibited form, potentially aiding in the protection and regulatory functions of telomeres. The model integrates our results with findings from other studies[36,38,71].

detected by MRN via a gating mechanism where ATP-dependent ring↔rod transitions of the CCDs allow loading of MR/MRN onto linear DNA, but not internal/circular DNA. In this model, ATP-dependent widening of the RAD50 CCDs enables DNA binding to the RAD50[NBD]'s. In case of binding near or at a DNA end, a DNA duplex would pass through the CCD ring, which can subsequently close to a rod with a channel for a single DNA duplex as observed in the structures. This mechanism could explain why MRN can load onto DNA ends sterically blocked by proteins and cleaves DNA at some distance from the terminus[45,69,70]. If circular DNA is bound, DNA would pass through the CCD ring twice, which might prevent CCD assembly and formation of the sensing state. Nevertheless, relation between CCD dynamics and DSB sensing requires further investigation.

Our results also uncover substantial species-specific differences between prokaryotic MR and human MRN. The two "resting states" of MR and MRN in the presence of ATPγS are highly similar, with open CCDs and autoinhibited Mre11 nuclease. However, an ATP hydrolysis step in the presence of DNA leads to a striking contrast between human MRNs and *Ec*MRs. Both complexes form a tight clamp around the DNA duplex with closed CCDs, but only in *Ec*MR, the MRE11 nuclease dimer relocates to the other side of DNA to form the "cutting state" conformation[37,40]. Evidently, in human MR/MRN, the MRE11 dimer remains in the auto-inhibited state with nuclease active sites blocked by RAD50, although the CCDs form a related rod-like conformation and both structures included an ATP hydrolysis step to form. We therefore refer to this new state of MRN/MR as "sensing" state to distinguish the DNA-bound state from the resting (ATP-bound but no DNA) and cutting states. Importantly, it solves the conundrum of how mammalian MRN on one hand can sense DNA ends through

CCD ring↔rod transitions, yet avoid automatic nuclease activation like in the bacterial system. Furthermore, the sensing state would be very well suited to adopt non-nuclease functions at DNA breaks such as the tethering or signaling functions of the MRN complex (Fig. 5). We do not yet know how the MRE11 dimer relocates to a putative cutting state in human MRN, but it is likely that phosphorylated CtIP, missing in our analysis, is a critical factor since it might stabilize the cutting state according to structure prediction and mutational analyses[71]. Based on the variability analysis and 2D classes, we note that apparently the MRE11 dimer is not fully fixed in the sensing state and can move with respect to the RAD50 dimer. It is plausible that ATP hydrolysis or repeated ATP cycles induce increased dynamics that enable ATM or CtIP to fully induce structural conformations that act in ATM activation and nuclease reactions. However, further studies are needed to reveal how the cutting state is induced.

Our structure also reveals some species-specific properties with respect to ATP hydrolysis and ATP analogs in stabilizing functional state and intermediates. In the sensing state of human MRN, the active sites of the best resolved structures contain density for ADP•BeF_x. A recently published *S. cerevisiae* MR-DNA complex bound by ATPγS has a similar "clamp" conformation with closed CCDs[72]. However, in our case, like in *Ec*MR, ATPγS stabilizes open CCDs and does not enable the formation of a DNA-bound clamp. Likewise, *Chaetomium thermophilum* MRN adopted closed CCDs in the presence of ATPγS and absence of DNA. It is unlikely that there are fundamental mechanistic differences with respect to the role of ATP binding and hydrolysis for MRN/X in different species. However, differences in interface strengths, in particular the CCD-CCD interaction, or physiological conditions in relation to cryo-EM might account for how well ATP

analogs recapitulate ATP binding and hydrolysis reactions. In any case, the precise role of ATP hydrolysis in DNA sensing needs further studies.

A single NBS1 subunit wraps around the MRE11 dimer, similar to what has been observed for fungal MRN[73]. The unexpected interaction of NBS1's C helix with RAD50, which was not observed in the fungal complex, explains a previously observed direct interaction between RAD50 and NBS1 in the absence of MRE11[74]. The interaction is intriguing, as it involves two important regulatory motifs of MRN that mediate interactions with other factors (ATM, TRF2, CtIP). The S site on RAD50 and the ATM recruitment motif on NBS1 bind to each other, shielding themselves from direct interaction with other factors. It restricts interactions with TRF2 to one side of the complex by blocking TRF2 binding to the occupied S site. AlphaFold 2/3 predictions of the interaction of iDDR with the S site were relatively accurate and extensive mutational studies have validated this interaction. The observed structure can also explain effects of early studies of RAD50 S site mutants in mice, where mice carrying the *Rad50*[K22M] allele (along a null allele) were viable, but exhibited telomere failures and ATM dependent apoptosis, while other MRN-associated activities were less affected[75,76]. K22[RAD50] forms a specific salt bridge with E468[TRF2], but not with NBS1. According to AlphaFold3 predictions, K22[RAD50] also forms a direct salt-bridge with CtIP, which may explain these observations with a preferential or stronger destabilization of the interaction with TRF2[iDDR].

Interestingly, TRF2 binds the sensing state without inducing significant conformational changes in CCDs, RAD50[NBD]s or MRE11. We also do not observe a stimulation of MRN's ATPase activity by TRF2 (Fig. 4E). A 2-3 fold stimulation of *S. cerevisiae* MR ATPase by Rif2 was observed in vitro, so there could be species-specific differences[55]. Furthermore, we do not observe that the presence of TRF2 and binding of iDDR prevents MRN from binding to DNA. It was previously observed that iDDR peptide reduces the interaction of isolated Rad50[NBD] with DNA. Instead of reduced binding, we obtained a saturated binding isotherm of MRN to DNA that is prebound by TRF2 (Fig. 4F), while binding of MRN alone does not reach saturation. A possible explanation is that interactions with TRF2 reduce clustering of MRN, which is a feature of both human[27] and yeast/fungal MR/MRX[68,73]. In any case, our observations argue against a model, where TRF2's function is to simply remove MRN from telomeres. Rather, it sequesters the sensing state at dysfunctional telomeres and might even stabilize it, consistent with TRF2-dependent co-localization of MR/MRN at telomeres[54], where MRN could adopt structural functions. Additional cell cycle-regulated interactions of TRF2's TRFH domain with NBS1 Y429QLSP433[77], prevent access of telomere-specific exonuclease Apollo in G1. These additional interactions, consistent with a 1:1 MRN-TRF2 complex, would strengthen the interaction between MRN and TRF2 and suggest that a segregated, nuclease-inhibited sensing state could help protect or assist with structural functions in dysfunctional G1 telomeres (Fig. 5).

Our structural results reveal that NBS1's C helix needs to dissociate from the RAD50 S site to from the MRN-ATM complex, with NBS1's C terminal region bound to ATM[65] (Fig. 5). An NBS1 variant (1-726, NBS1[ΔC]) that lacks the RAD50 interacting C helix resulted in a hypomorphic phenotype with most MRN functions intact, but improper phosphorylation of a subset of ATM targets such as SMC1[64], while the deletion also mitigated the effect of the RAD50S mutation. Since the C-terminal motif is not sufficient for activation of ATM by MRN[64] and other parts of the complexes are needed as well, it is plausible that detachment of the NBS1 C helix on one hand, leads to a more stable MRN-ATM interaction, whereby, similar to MRX-Tel1[78], other parts of MRN are bound by ATM as well. Along the same line, the C helix would likely need to detach from the S site to enable MRE11 to relocate to the other side of DNA to form the cutting state. However,

since deletion of parts of the C helix did not lead to increased genome instability in mice[64], it might not play a critical role in the cutting state.

In summary, we identify a DSB sensing state of MRN and MRN-TRF2 that argues for the DNA end chemistry insensitive detection of DSBs through a gating mechanism that can load MRN onto loose DNA ends, whereby the S site of Rad50 emerges as structural hub for both intrinsic and extrinsic regulators (Fig. 5). The structural analysis solves the conundrum how DNA end detection and nuclease functionalities are separated in eukaryotes and mammals and provide a framework to understand subsequent regulatory steps. In the future, the precise roles of ATP hydrolysis in DNA end processing, nuclease activities, and what structural transitions happen to form the cutting state and the ATM complex of MRN need to be resolved. Further work is also needed to understand a possible mechanistic function of a TRF2-bound sensing state at dysfunctional telomeres.

## Methods

### Organisms as source for materials used in experiments
For the amplification of plasmid DNA, we used *Escherichia coli* XL1 Blue cells and *E. coli* DH10MultiBac cells for bacmid generation. *Spodoptera frugiperda* (Sf21) insect cells were used for virus production. *E. coli* Rosetta2, *Trichoplusia ni* insect cells (Hi5) and *Homo sapiens* Expi293F cells were used for recombinant protein expression.

### Expression and purification of MR and MRN complexes
MRN complexes were either expressed in Expi293F cells or Hi5 cells. Purification protocols were adapted from our previously described protocols and applied to purifications from both expression systems[73,79]. In brief: pACEMam1_pMDC_M[H129N]RN plasmid was transfected into 500 mL Expi293F cells, cultured for 72 h at 37 °C and harvested by centrifugation (1000 g, 4 °C, 10 min). In case when insect cells were used as the expression system, bacmids were generated in *E. coli* DH10MultiBac as previously described according to the Multi-Bac system by transformation into *E. coli* DH10MultiBac cells followed by blue-white selection[80]. Baculoviruses containing wild-type MR or MRN were generated in Sf21 cells. Virus titers were determined by small-scale test expressions. 1 L of Hi5 cells (Invitrogen) seeded to $1 \times 10^6$ cells/mL were infected with respective baculovirus and cultured for 72 h at 27 °C. Cells were harvested by centrifugation (1,900 g, 4 °C, 15 min). All subsequent purification steps were carried out either in the cold room or on ice. Pellets were homogenized in lysis buffer (50 mM HEPES pH 8.0, 250 mM NaCl, 10 % glycerol, 0.1 mM DTT, 0.5 mM MnCl$_2$, 1 mM MgCl$_2$, two complete EDTA-free Protease Inhibitor Cocktail tablets (Roche), 2 μL TURBO DNase I (Thermo Fisher Scientific), 20 mM NaF, and 12 mM Na$_2$P$_2$O$_7$). Lysates were sonicated for 2 × 45 sec (40 % duty cycle, 5 output control) and centrifuged at 30,000 g for 1 h at 4 °C. Supernatants were filtered through a Millex® glass fiber filter unit, Millex®-AP 50 (Roth). Clarified lysates were applied onto 2 mL FLAG M2 affinity resin (Sigma) in a gravity flow column, equilibrated in low salt buffer (50 mM HEPES pH 8.0, 250 mM NaCl, 10 % glycerol, 0.1 mM DTT, 0.5 mM MnCl$_2$) and incubated for 2 h rolling. Beads were washed with 10 column volumes (CV) of low salt buffer, 5 CV of high salt ATP-buffer (50 mM HEPES pH 8.0, 1 M NaCl, 2 mM ATP, 5 mM MgCl$_2$ 10 % glycerol, 0.1 mM DTT, 0.5 mM MnCl$_2$) followed by 5 CV of high salt buffer w/o ATP (50 mM HEPES pH 8.0, 1 M NaCl, 10 % glycerol, 0.1 mM DTT, 0.5 mM MnCl$_2$). Afterwards, beads were re-equilibrated in low salt buffer. Protein was eluted with 0.2 mg/mL 3x FLAG-peptide (TargetMol) in low salt buffer to a total volume of 5 CV. The eluate was concentrated using an Amicon® Ultra Centrifugal Filter, 100 kDa MWCO and applied to size exclusion chromatography (Superose 6, 10/300, Cytiva) in SEC buffer (25 mM HEPES pH 8.0, 150 mM NaCl, 1 mM DTT). Stoichiometric MRN complex eluted at 0.5 CV. The integrity of the peak fractions was analyzed by SDS-PAGE (SERVA) and confirmed by Mass Photometry (Refeyn), for which

 

proteins were diluted to 50 nM just before the measurement (Supplementary Fig. 1A, B).

## Expression and purification of TRF2 complexes

The full-length human TRF2 gene was amplified from Addgene plasmid #185449 shared by Ahmet Yildiz (http://n2t.net/addgene:185449; RRID:Addgene_185449) and subcloned into a pFastBac vector using NheI and XhoI (pFB-MBP-3C-*Hs*TRF2-His_{10}). Baculovirus containing TRF2 was generated according to the protocol described for MR/MRN. Expression of full-length TRF2 was carried out in 1 L of Hi5 insect cells. Cells were harvested by centrifugation (500 *g*, 4 °C, 15 min) 72 h post-infection, flash-frozen in liquid nitrogen, and stored at −80 °C.

For purification, 10 mL of cell pellets were thawed and resuspended in 70 mL of lysis buffer (50 mM Tris-HCl, pH 7.5, 1 mM TCEP) supplemented with 1 mM phenylmethylsulfonyl fluoride (PMSF) and one tablet of Sigma protease inhibitor cocktail. The cells were allowed to swell for 20 minutes at 4 °C with gentle stirring. Subsequently, 40 mL of 50 % glycerol and 7.8 mL of 5 M NaCl were added dropwise while mixing, followed by a 30-minute incubation. The cell suspension was centrifuged at 38,000 g for 45 min at 4 °C to obtain clear lysate, which was then filtered through a Millex® glass fiber filter unit, Millex®-AP 50 (Roth).

The resulting supernatant, supplemented with 10 mM imidazole, was incubated with 1 mL of pre-equilibrated Ni-NTA agarose resin (Machery-Nagel) for 1 h at 4 °C. The resin was washed with 100 CV of high-salt buffer (lysis buffer + 500 mM NaCl + 20–50 mM imidazole) followed by 20 mL of low-salt buffer (lysis buffer + 100 mM NaCl). MBP-TRF2-His_{10} was eluted using 15 CVs of low-salt buffer with 500 mM imidazole. To remove the MBP tag, PreScission protease (1 µg per 50 µg MBP-TRF2-His_{10}) was added to the eluate and incubated for 3 h at 4 °C. The TRF2 protein was further purified using a 1 mL HiTrap Heparin column (Cytiva), developed with a NaCl gradient (100–1000 mM) in lysis buffer containing 0.5 mM TCEP and 5 % glycerol. Fractions containing full-length TRF2 were pooled and concentrated to approximately 1 mL before being injected onto a Superdex 200 Increase 10/300 GL column (Cytiva) equilibrated in storage buffer (25 mM HEPES, pH 7.5, 150 mM NaCl, 0.5 mM TCEP, and 5 % glycerol). Purity was assessed via SDS-PAGE. Peak fractions were pooled, concentrated to approximately 12 µM (dimer), aliquoted, and flash-frozen in liquid nitrogen.

TRF2 variants, including TRF2^{iDDR} (438-485) and TRF2^{iDDR-Myb} (438-542), were expressed as His_6-SUMO fusion proteins in *E. coli* Rosetta cells using pET47b vectors. Transformed Rosetta cells were grown in 4 L of 2YT medium (2× Yeast Extract Tryptone) supplemented with kanamycin (50 µg/mL) and chloramphenicol (30 µg/mL) at 37 °C with aeration until the culture reached an OD_{600} of approximately 1.0. Protein expression was induced by adding 0.5 mM IPTG, followed by incubation at 16 °C for 18 h. Cells were harvested (5000 g, 4 °C, 15 min), flash-frozen in liquid nitrogen, and stored at −20 °C.

For purification, the cell pellet was resuspended in 100 mL of lysis buffer supplemented with 500 mM NaCl, 1 mM PMSF, and one Sigma protease inhibitor cocktail tablet. Cells were lysed via sonication, and the lysate was clarified by centrifugation (38,000 g, 45 min, 4 °C). The supernatant, supplemented with 10 mM imidazole, was applied via gravity flow to 6 mL of pre-equilibrated Ni-NTA agarose resin (Machery-Nagel). The column was washed with 15 CV of high-salt buffer (lysis buffer + 1 M NaCl + 20 mM imidazole), followed by 8 CV of low-salt buffer (lysis buffer + 100 mM NaCl + 40 mM imidazole). The TRF2 variants were eluted using 3 CV of low-salt buffer supplemented with 500 mM imidazole. To remove the His_6-SUMO fusion tag, GST-tagged PreScission protease was added to the eluate (1:50 enzyme-to-protein ratio), and the mixture was dialyzed overnight at 4 °C against 1 L of buffer (50 mM Tris-HCl, pH 7.5, 100 mM NaCl, 1 mM TCEP, and 1 mM EDTA). The His_6-SUMO tag and uncleaved fusion protein were removed via a 6 mL Ni-NTA column, while PreScission protease was

eliminated using a 2 mL Glutathione Sepharose 4 Fast Flow column. Final purification was performed using a Superdex 200 Increase 10/300 GL column (Cytiva) equilibrated in storage buffer (25 mM HEPES, pH 7.5, 150 mM NaCl, 1 mM TCEP, and 1 mM EDTA). Protein purity was confirmed via SDS-PAGE. Peak fractions were pooled, concentrated to approximately 417 µM, aliquoted, and flash-frozen in liquid nitrogen.

## DNA substrates

The oligonucleotides used in this study were acquired from Metabion (Germany). Modified sequences (6-FAM labeled) were obtained HPLC-purified and lyophilized. Complementary ssDNA sequences were annealed to one another in a PCR cycler starting at 95 °C with gradually decreasing temperatures (0.1 °C/sec.) down to 25 °C. For annealing, an annealing buffer was used (25 mM TRIS pH 7.5, 50 mM NaCl, 10 mM MgCl_2). In case of modified sequences, the complementary sequence was used in 1.1x excess compared to the labeled/modified oligonucleotide. dsDNA was either immediately used or stored at −20 °C.

## Fluorescence anisotropy DNA binding assays

Fluorescence polarization anisotropy (FA) was used to monitor MR/MRN and TRF2 binding to 64 bp Myb DNA (see Table Oligonucleotides). 30 nM of 6-FAM-labeled DNA was incubated with increasing amounts of MRN and/or TRF2/MRN in assay buffer (25 mM HEPES 8.0, 100 mM NaCl, 5 mM MgCl_2, 0.2 mg/ml BSA, 1 mM DTT). 1 mM ATP was added if not stated otherwise. When both MRN and TRF2 present in the reaction, a constant concentration of TRF2 (1.6 µM) was used with increasing amounts of MRN. FA was measured at an excitation wavelength of 488 nm and an emission wavelength of 520 nm using an automated polarization microscope[81,82]. Measurements were taken every 5 min for 30 min to monitor MRN-DNA binding stability. In each well, twelve different z-planes were measured to reduce the possibility of erroneous FA values caused by potential fluorescing protein-DNA aggregates. The reactions were prepared and measured in triplicates. The median FA values were calculated as described in ref. 73. The final FA value of each sample was the average of median FA values over the time course of 30 min.

## NADH-coupled ATPase assay

ATPase rates of MR/MRN complex were assessed using a plate-reader-based ATPase assay, that is coupled to NADH oxidation. All steps prior to the measurement were carried out on ice. 250 nM protein was incubated with assay buffer (100 mM NaCl, 25 mM HEPES pH 7.5, 2 mM MgCl_2, 1 mM MnCl_2, 1 mM phosphoenolpyruvate (PEP), 0.15 mM ATP, 25 U/mL Lactate Dehydrogenase/Pyruvate kinase (Sigma-Aldrich), 0.1 mM NADH, 0.1 mg/mL BSA (NEB), 0.5 mM DTT) in 50 µL final reaction volume. To stimulate ATPase activity, 64 bp Myb DNA was added to achieve 1000 nM final DNA concentration. TRF2^{iDDR-Myb} was added in 5-fold molar excess to DNA where stated. Decreasing NADH concentrations correlate with fluorescence decay of NADH which was monitored up to 4 hours at 37 °C in a flat-bottom, non-binding black 384-well plate (Greiner Bio-One) using 340 nm laser for excitation and emission at 460 nm in a Tecan Spark plate reader. The reactions were prepared and measured in triplicates. ATP turnover was determined using steady-state rate at maximal initial linear rates and corrected subtracting buffer blank.

## Cryo-EM grid preparation

Freshly purified MR/MRN was directly used for cryo-EM sample preparation. The proteins were diluted to final concentrations of 0.4 µM (-0.2 mg/mL) in 25 mM HEPES pH 7.5, 120 mM NaCl, 5 mM MgCl_2, 1 mM MnCl_2, 1 mM DTT, 1 mM ATP and incubated for 10 min at room temperature. 50 bp DNA substrate (see Table Oligonucleotides) was added to reach a final concentration of 0.3 µM and incubated at 35 °C for at least 30 min followed by the addition of 1 mM BeF_x. In case TRF2 full-length or TRF2^{iDDR-Myb} was added, 64 bp Myb DNA and TRF2 were pre-

incubated for 10 min at 35 °C, then added to the MR(N) complex. Grids were prepared using a Leica EM GP plunge freezer (Leica) at 12 °C and 95% humidity. Just before plunging, the reaction was supplemented with octyl β-D-glucopyranoside (β-OG) at a final concentration of 0.05%. 4.5 μL of sample were applied onto a glow-discharged Quantifoil® R2/1 + 2 nm carbon Cu 200 grid. The samples were pre-blotted for 20 sec, blotted for 2.3 s, before vitrification in liquid ethane.

## Cryo-EM data acquisition collection

MR-DNA dataset was collected on FEI Titan Krios G3 transmission electron microscope (300 kV) equipped with a GIF quantum energy filter (slit width 20 eV) and a Gatan K2 Summit direct electron detector (software used: EPU 3.5.1, TEM User interface Titan 3.15.1, Digital Micrograph 3.22.1461.0). All other datasets were collected on FEI Titan Krios G3 transmission electron microscope (300 kV) equipped with a Selectris X imaging filter (slit width 10 eV) and a Falcon 4 direct electron detector (software used: EPU 3.5.1, TEM User interface Titan 3.15.1).

For the MR-DNA dataset, 4047 movies were collected at a nominal magnification of 130,000x (1.06 Å/pix), a defocus range of −1.1 to −2.9 μm, and a total electron dose of 42.377 e⁻/Å² fractionated into 40 frames over 10 sec. The MRN-DNA dataset (16,890 movies), MRN-ATPγS dataset (7937 movies), MR-TRF2$^{iDDR·Myb}$-DNA dataset (18,134 movies), MRN-TRF2$^{iDDR·Myb}$-DNA dataset (33,072 movies), MRN-TRF2-DNA dataset (39,320 movies) were collected at a nominal magnification of 165,000x (0.727 Å/pix), a defocus range of −0.5 to −2.6 μm, and a total electron dose of 40 e⁻/Å² fractionated into 40 movie frames over 2.75 sec.

## Cryo-EM image processing

Movie frames were motion corrected using MotionCor2 1.4.5[83]. All subsequent cryo-EM data processing steps were carried out using CryoSPARC (4.6.2 and former versions). The CTF parameters of the datasets were determined using patch CTF estimation job integrated in the CryoSPARC software framework[84]. The masks for the calculation of FSC were created by CryoSPARC automasking. The resolutions reported are calculated based on the gold-standard Fourier shell correlation criterion (FSC = 0.143) with 3D FSC plots using Remote 3DFSC Processing Server[85]. The exact processing schemes are depicted in Supplementary Fig. 2, 3, 7, 9, 10, and 11. Data collection and refinement statistics are summarized in Table 1.

For the MR-DNA structure, particles were initially picked on 4047 micrographs using a blob picker with an applied diameter of 120 to 360 Å (Supplementary Fig. 2). Picked particles were extracted with a box size of 360 pixels, and subjected to multiple rounds of 2D classification, resulting in an initial cleaned subset of 2D classes comprising 225,632 particles. The 2D classes with clearly defined features were selected and served as templates for a reference-based template picking approach, and subsequently as input for Topaz train[86,87]. A total pool of 817,041 particles was further sorted by ab initio reconstruction and heterogeneous refinement. The yielded class of 301,918 particles with clearly defined features was further sorted by rounds of 2D classification, homogeneous refinement and non-uniform refinement, leaving a new particle set of 226,766 particles. 3D classification was applied to separate conformational states of MR with symmetrically attached MRE11 and partially detached MRE11. The final resolution of MR-DNA reconstruction (95,397 particles), after homogeneous refinement and non-uniform refinement of the former class, was 3.21 Å. The sharpened map of this reconstruction was used further for model building and refinement.

For the MRN-DNA structure, particles were initially picked on 16,890 micrographs using a blob picker with an applied diameter of 120 to 360 Å (Supplementary Fig. 3). Picked particles were extracted with a box size of 360 pixels, and subjected to multiple rounds of 2D classification, resulting in an initial cleaned subset of 2D classes

comprising 573,534 particles. Reasonable 2D classes were selected and served as templates for a reference-based template picking approach, and subsequently as input for Topaz train[86,87]. A total pool of 929,509 particles was further sorted by 2D classification to remove a subset of 206,194 poorly resolved particles. The remaining 723,315 particles were subjected to ab-initio reconstruction and heterogeneous refinement. Particles in reasonable 3D volumes were further sorted by homogeneous refinement and 3D classification, which allowed separation of two well-resolved states of MRN-DNA: MRN with symmetrically attached MRE11 (175,312 particles) and partially detached MRE11 (204,607 particles). 3D classification and homogeneous refinement were further applied to the former class, leading to the final MRN-DNA reconstruction at 3.08 Å using 172,784 particles. The sharpened map of this reconstruction was used further for model building and refinement.

For the MRN-ATPγS map, particles were initially picked on 7937 micrographs using blob picker with an applied diameter of 120 to 360 Å (Supplementary Fig. 7). Picked particles were extracted with a box size of 360 pixels, and subjected to multiple rounds of 2D classification, resulting in an initial cleaned subset of 2D classes comprising 127,848 particles. Reasonable 2D classes were selected and served as templates for a reference-based template picking approach, and subsequently as input for Topaz train[86,87]. A total pool of 308,429 particles was further sorted by 2D classification to remove a subset of 139,004 poorly resolved particles. The remaining 169,425 particles were subjected to ab-initio reconstruction and heterogeneous refinement. Particles in the more reasonable 3D volume were further sorted by rounds of 2D classification, homogeneous refinement and non-uniform refinement, leading to the final MRN-ATPγS reconstruction at 3.27 Å using 75,330 particles. AlphaFold3[63] predicted model of MRN-ATP was rigid-body-fitted into the map without much further adjustment and was used for the preparation of the figures.

For the MR-TRF2$^{iDDR·Myb}$-DNA structure, particles were initially picked on 18,134 micrographs using a blob picker with an applied diameter of 120 to 360 Å (Supplementary Fig. 9). Picked particles were extracted with a box size of 360 pixels, and subjected to multiple rounds of 2D classification, resulting in an initial cleaned subset of 2D classes comprising 599,266 particles. Reasonable 2D classes were selected and served as templates for a reference-based template picking, and subsequently as input for Topaz train[86,87]. A total pool of 1,109,607 particles were subjected homogeneous refinement and 3D classification, which allowed separation of two well-resolved states of MRN-DNA: MRN with symmetrically attached MRE11 (369,251 particles) and partially detached MRE11 (319,438 particles). Particles from the former class were further sorted by homogeneous refinement, non-uniform refinement and 3D classification. The final resolution of MR-TRF2$^{iDDR·Myb}$-DNA reconstruction after non-uniform refinement was 2.7 Å containing 274,928 particles. The sharpened map of this reconstruction was used further for model building and refinement. Particles from the MRE11-partially-detached class were further processed by 2D and 3D classification, followed by iterative rounds of homogeneous refinement. The resulting subset of 203,202 particles was used to reconstruct the cryo-EM map of MR-TRF2$^{iDDR·Myb}$-DNA (MRE11-detached). The sharpened map of this reconstruction was used further for model building and refinement. This map and model were only used for the preparation of the figures.

For the MRN-TRF2$^{iDDR·Myb}$-DNA structure, particles were initially picked on 33,072 micrographs using a blob picker with an applied diameter of 120 to 360 Å (Supplementary Fig. 10). Picked particles were extracted with a box size of 360 pixels, and subjected to multiple rounds of 2D classification, resulting in an initial cleaned subset of 2D classes comprising 429,260 particles. Reasonable 2D classes were selected and served as templates for a reference-based template picking, and subsequently as input for Topaz train[86,87]. A total pool of

**Table 1 | Data collection, refinement and validation statistics**

| | MR-DNA (EMDB-52959) (PDB 9Q9H) | M$^{H129N}$RN-DNA (EMDB-52960) (PDB 9Q9I) | M$^{H129N}$RN-ATPγS (EMDB-54397) | MR-TRF2$^{iDDR-Myb}$-DNA (EMDB-52962) (PDB 9Q9K) | MRN-TRF2$^{iDDR-Myb}$_-DNA (EMDB-52961) (PDB 9Q9J) | MRN-TRF2-DNA (EMDB-52964) (PDB 9Q9M) |
|---|---|---|---|---|---|---|
| **Data collection and processing** | | | | | | |
| Magnification | 130,000 | 165,000 | 165,000 | 165,000 | 165,000 | 165,000 |
| Voltage (kV) | 300 | 300 | 300 | 300 | 300 | 300 |
| Electron exposure (e⁻/Å²) | 42.377 | 40 | 40 | 40 | 40 | 40 |
| Defocus range (µm) | -1.1 to -2.9 | -0.5 to -2.6 | -0.5 to -2.6 | -0.5 to -2.6 | -0.5 to -2.6 | -0.5 to -2.6 |
| Pixel size (Å) | 1.06 | 0.727 | 0.727 | 0.727 | 0.727 | 0.727 |
| Symmetry imposed | C1 | C1 | C1 | C1 | C1 | C1 |
| Initial particle images (no.) | 817,041 | 723,315 | 169,425 | 1,109,607 | 725,737 | 1,155,493 |
| Final particle images (no.) | 95,397 | 172,784 | 75,330 | 274,928 | 129,300 | 208,916 |
| Map resolution (Å) | 3.21 | 3.08 | 3.27 | 2.70 | 3.01 | 2.97 |
| FSC threshold | 0.143 | 0.143 | 0.143 | 0.143 | 0.143 | 0.143 |
| **Refinement** | | | | | | |
| Initial model used (PDB code) | AlphaFold3 | AlphaFold3 | | 9Q9H | 9Q9I | 9Q9J |
| Model resolution (Å) | 3.2 | 3.2 | | 2.9 | 3.1 | 3.2 |
| FSC threshold | 0.5 | 0.5 | | 0.5 | 0.5 | 0.5 |
| Map sharpening $B$ factor (Å²) | -110 | -85 | | -89 | -87 | -80 |
| Model composition | | | | | | |
| Non-hydrogen atoms | 16,786 | 17,121 | | 16,910 | 17,465 | 17,361 |
| Protein residues | 1932 | 1960 | | 1934 | 2004 | 1993 |
| Nucleotides | 50 | 54 | | 54 | 52 | 52 |
| $B$ factors (Å²) | | | | | | |
| Protein | 62.66 | 110.93 | | 60.01 | 78.74 | 78.55 |
| Nucleotide | 99.03 | 93.51 | | 46.38 | 52.12 | 52.15 |
| Ligand | 24.04 | 92.97 | | 34.21 | 54.76 | 54.76 |
| Water | 24.41 | 82.39 | | 28.03 | 44.38 | 44.38 |
| RMSDs | | | | | | |
| Bond lengths (Å) | 0.006 | 0.006 | | 0.006 | 0.006 | 0.006 |
| Bond angles (°) | 1.203 | 1.161 | | 1.161 | 1.076 | 1.174 |
| Validation | | | | | | |
| MolProbity score | 2.09 | 2.02 | | 1.93 | 1.71 | 1.95 |
| Clashscore | 10.77 | 10.65 | | 10.48 | 8.18 | 10.22 |
| Poor rotamers (%) | 0.8 | 0.7 | | 0.5 | 0.4 | 0.5 |
| Ramachandran plot | | | | | | |
| Favored (%) | 95.77 | 95.92 | | 96.81 | 97.53 | 96.65 |
| Allowed (%) | 4.23 | 4.08 | | 3.19 | 2.47 | 3.35 |
| Disallowed (%) | 0.00 | 0.00 | | 0.00 | 0.00 | 0.00 |

725,737 particles were subjected homogeneous refinement and 3D classification, which allowed separation of two well-resolved states of MRN-TRF2$^{iDDR-Myb}$-DNA: MRN with symmetrically attached MRE11 (292,228 particles) and partially detached MRE11 (146,293 particles). Particles from the former class were further sorted by homogeneous refinement and 3D classifications, followed by non-uniform refinement. The final resolution of MRN-TRF2$^{iDDR-Myb}$-DNA reconstruction after homogeneous refinement was 3.01 Å (129,300 particles). The sharpened map of this reconstruction was used further for model building and refinement.

For the MRN-TRF2-DNA structure, particles were initially picked on 39,320 micrographs using a blob picker with an applied diameter of 120 to 360 Å (Supplementary Fig. 11). Picked particles were extracted with a box size of 360 pixels, and subjected to multiple rounds of 2D classification, resulting in an initial cleaned subset of 2D classes comprising 435,078 particles. Reasonable 2D classes were selected and served as templates for a reference-based template picking, and subsequently as input for Topaz train[86,87]. A total pool of 1,155,493 particles were subjected homogeneous refinement and 3D classification, which allowed separation of two well-resolved states of MRN-TRF2-DNA: MRN with symmetrically attached MRE11 (553,435 particles) and partially detached MRE11 (159,492 particles). Particles from the former class were further sorted by rounds of homogeneous refinement and 3D classification. The resulting subset of 208,916 particles was subjected to a last round of non-uniform refinement, and used to reconstruct the cryo-EM map of MRN-TRF2-DNA at 2.97 Å. The sharpened

map of this reconstruction was used further for model building and refinement.

## Model building

All EM structures discussed in this study were constructed based on an atomic model of the MRN-DNA complex predicted by AlphaFold3[63]. This model was initially docked as a rigid body into the refined 3D reconstruction maps using ChimeraX 1.6.1[88], followed by manual adjustments and real-space refinement in COOT 0.9.8.1[89] to achieve the best fit to the density. The bound TRF2 fragment was built de novo, guided by prominent side chains and secondary structure predictions from AlphaFold3[63]. Due to map quality limitations, the double-stranded DNA sequences in all structures were modeled as AT-pair oligos. The presence of $BeF_x$ and ADP in the nucleotide-binding pocket of RAD50 was confirmed by well-defined densities in the highest-resolution EM map. The models underwent iterative cycles of real-space refinement and manual adjustments using Phenix 1.20.1-4487[90] and COOT 0.9.8.1[89], resulting in excellent stereochemistry as validated by MolProbity. All figures were generated using PyMOL (The PyMOL Molecular Graphics System, 2.5.5, Schrödinger, LLC) and ChimeraX 1.6.1[88].

## Crosslinking mass spectrometry

MRN (produced in Expi293F cells, variant MR$^{apex}$N) was used for crosslinking using BS3 (bis(sulfosuccinimidyl)suberate)) crosslinker (Thermo Fisher Scientific). MRN was incubated with reaction buffer to achieve final concentrations of 300 nM protein and 500 nM 80 bp dsDNA in 25 mM HEPES pH 7.5, 100 mM NaCl, 5 mM $MgCl_2$, 1 mM $MnCl_2$, 1 mM ATP, 1 mM DTT) and ATM (purified according to ref. 79) was added to the MRN complex for 30 min at 30 °C. BS3 crosslinker was reconstituted freshly according to the manufacturers protocol (147 μM final BS3 in the sample) and incubated for 20 min. The reaction was quenched with 1 mM Tris-HCl pH 7.5, supplemented with 4x SDS dye and analyzed on a SERVAGel™ TG PRiME™ 4–12 %. The gel was stained with Coomassie dye.

## In-gel digestion

The excised Coomassie-stained gel (one sample) was cut into small cubes, followed by destaining in 50 % ethanol/25 mM ammonium bicarbonate. Proteins were then reduced in 10 mM DTT at 56 °C and alkylated by 50 mM iodoacetamide in the dark at room temperature. Afterwards, proteins were digested by trypsin (1 μg per sample) in 50 mM ammonium bicarbonate at 37 °C overnight. Following peptide extraction sequentially in 30 % and 100 % acetonitrile, the sample volume was reduced in a centrifugal evaporator to remove residual acetonitrile. The peptides were then acidified with formic acid and purified via solid phase extraction in $C_{18}$ StageTips[91].

## Liquid chromatography tandem mass spectrometry

Peptides were analyzed on an Orbitrap Exploris 480 mass spectrometer (Thermo Fisher Scientific) coupled to an EASY-nLC 1200 UHPLC system (Thermo Fisher Scientific). Peptides were separated in an in-house packed 55-cm analytical column (inner diameter: 75 μm; ReproSil-Pur 120 C18-AQ 1.9-μm silica particles, Dr. Maisch GmbH) by online reversed phase chromatography through a 90 min gradient of 2.4–32 % acetonitrile with 0.1 % formic acid at a nanoflow rate of 250 nL/min. The eluted peptides were sprayed directly by electrospray ionization into the mass spectrometer. Each sample was injected twice and measured using two different combinations of collision energies in stepped mode[92]. Mass spectrometry was conducted in data-dependent acquisition mode using a top10 method with one full scan (resolution: 60,000, scan range: 300-1650 m/z, target value: 3 × 10$^6$, maximum injection time: 60 ms) followed by 10 fragment scans via higher energy collision dissociation (HCD; normalized collision energy in stepped mode: 25, 30, 35 % or 27, 30, 33 %; resolution: 30,000, target value: 1 × 10$^5$, maximum injection time: 60 ms, isolation window: 1.4 m/z). Only precursor ions of +3 to +8 charge state were selected for fragment scans. Additionally, precursor ions already isolated for fragmentation were dynamically excluded for 25 s.

## Mass spectrometry data analysis

Raw data files were pre-processed using the MaxQuant software (version 2.1.3.0)[93] as previously described[94] ignoring the de-noising filter. The peak lists (*.HCD.FTMS.sil0.apl files) were searched using xiSEARCH (version 1.8.7)[95] against a target-decoy database consisting of the protein sequences of the MRN complex and the ATM. Trypsin/P specificity was assigned. Up to 4 missed cleavages were allowed. Crosslink search was based on the BS3 specificity linking K, S, T, Y residues and the protein N-terminus. Carbamidomethyl on cysteine was assigned as fixed modification. Variable modifications included methionine oxidation, BS3 mono-link with a hydrolyzed or Tris-quenched end and loop link. Mass tolerance was 5 ppm at the MS1 level and 6 ppm at the MS2 level. Residue pairs-level FDR was set to 1%.

## Visualization of Crosslinks

The processed data and the coordinates of the MRN-DNA structure were imported to XiVIEW[96] to visualize crosslinks among the MRE11, RAD50, NBS1 subunits by mapping the links onto the EM-structure. ATM links have been omitted from the map. The False Discovery Rate (FDR) was set to 1 % and the crosslinking distance cutoff to <30 Å. The match score was adjusted to a minimum 10 and maximum 20. The circular crosslinking map was generated in XiVIEW along with the structural map comprising the crosslinks, which was exported to ChimeraX[88].

## Quantification and statistical analysis

To analyze ATPase rates, the linear fluorescence decrease was fitted to a linear regression in Prism 10.4.0 (GraphPad) and the slope was used to determine the ATPase rate. Equation (1) was used for the exact calculation of data in Fig. 1B and Fig. 4E.

$$ATPase\ rate[ATP\ complex^{-1}s^{-1}]$$
$$= \frac{slope[FUs^{-1}]}{NADH\ slope\left[-261.4\frac{FU}{\mu M\ ATP}c_{MR(N)}[\mu M\ complex]\right]} \tag{1}$$

The NADH slope was calculated from a calibration curve that was recorded by carrying out a titration of varying ADP concentrations ranging from 0 to 100 μM in 10 μM increments to constant NADH concentration in the solution lacking protein and DNA.

Direct fluorescence anisotropy data of MRN binding to 64 bp Myb DNA was analyzed using Prism 10.4.0 (GraphPad) by fitting the anisotropy data to a One site Specific binding with Hill slope model, where Eq. (2) was employed (Fig. 2C, Fig. 4F):

$$FU = \frac{B_{max}[MRN]^h}{K_D{}^h + [MRN]^h} \tag{2}$$

Direct fluorescence anisotropy data of TRF2 binding to 64 bp Myb DNA was analyzed using Prism 10.4.0 (GraphPad) by fitting the anisotropy data to a Two sites Specific binding model, where Eqs. (3)–(5) were employed (Supplementary Fig. 12A):

$$FU(site1) = \frac{B_{max}Hi * [TRF2]}{K_DHi + [TRF2]} \tag{3}$$

$$FU(site2) = \frac{B_{max}Lo * [TRF2]}{K_DLo + [TRF2]} \tag{4}$$

$$FU(total) = FU(site1) + FU(site2) \tag{5}$$

## Reporting summary

Further information on research design is available in the Nature Portfolio Reporting Summary linked to this article.

## Data availability

Cryo-EM data generated in this study were deposited as coordinate files in the Protein Data Bank (https://www.wwpdb.org/) and reconstructions in the Electron Microscopy Data Bank (https://www.ebi.ac.uk/emdb/) under the following accession codes for each structure: MR-DNA complex: EMDB: 52959 (3.21 Å) and PDB: 9Q9H. $M^{H129N}RN$-DNA complex: EMDB: 52960 (3.08 Å) and PDB: 9Q9I. $M^{H129N}RN$-ATPγS complex: EMDB: 54397 (3.27 Å). MR-TRF2$^{iDDR-Myb}$-DNA complex: EMDB: 52962 (2.7 Å) and PDB: 9Q9K. MRN-TRF2iDDR-Myb-DNA complex: EMDB: 52961 (3.01 Å) and PDB: 9Q9J. MRN-TRF2-DNA complex: EMDB: 52964 (2.97 Å) and PDB: 9Q9M. Crosslinking-mass spectrometry data are available via ProteomeXchange with identifier PXD067324. Plasmids and materials that were generated in this study as well as further information requests are available from the corresponding author. Source data generated and analyzed during this study are available in the Zenodo repository (https://doi.org/10.5281/zenodo.16914563). Source data are provided with this paper.

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

## Acknowledgements

We thank the Cryo-EM Core Facility of the Gene Center, Department of Biochemistry, LMU Munich for technical support during EM data collection and all members of the Hopfner lab for helpful discussions. We are grateful to Olga Fettscher, Brigitte Kessler, Halina Kurzyca-Börncke, Manuela Moldt and Alexandra Scheele for excellent technical assistance. We thank Mauro Modesti (Cancer Research Center of Marseille) for the gift of Expi293F cells that were used for protein expression. Y.F. and F.K. acknowledge support from the International Max-Planck Research School for Molecules of Life and F.K. additionally thanks the Elite network of Bavaria/ENB Biological physics program. This work was supported by the Deutsche Forschungsgemeinschaft (CRC1361, Gottfried Wilhelm Leibniz-Prize, HO2489/11-1 to K.P.H.). Funding of CRC1361 supported the Exploris 480 system (J.X.C.).

## Author contributions

Y.F. and F.K. conducted all experiments (MR and MRN purification, cryo-EM sample preparation, in vitro biochemistry assays, chemical cross-linking) and together with M.K. and K.L. collected cryo- EM data. Y.F., F.K., H.C., and K.P.H. processed EM-data, with advice from M.K. H.C. purified TRF2 variants and carried out structural model building, refinement, and deposition. J.X.C. performed CX-MS analysis. C.J. performed fluorescence anisotropy data collection. Y.F., F.K., H.C., K.P.H., and K.L. analyzed and interpreted data. K.P.H. designed and supervised the overall project and provided funding. Y.F., F.K., H.C., K.L., and K.P.H. wrote the manuscript with input from all authors.

## Funding

## Competing interests

The authors declare no competing interests.
