## [Transparent Peer Review file · Nature Communications]

Structural Basis for DNA Break Sensing by Human MRE11-RAD50-NBS1 and its Regulation by Telomeric Factor TRF2

Corresponding Author: Professor Karl-Peter Hopfner

Version 0:

Reviewer comments:

Reviewer #1

(Remarks to the Author)

Fan et al determine cryo-EM structures of the human MR and MRN complexes with DNA and the Shelterin component TRF2. This stunning work reveals the complex molecular logic behind the regulation of MRN nuclease activity at telomeres. Overall, the work provides a clear molecular explanation for decades of work from multiple labs on the regulation of human MRN activities, and how this layer of complexity differs from bacterial MR orthologs. The paper is clearly written and illustrated, and near ready for publication already. I have minor points.

Points:

1. The use of the “DNA topology” and “topological sensing” throughout the text is confusing in the context of how we normally define DNA topology in terms of supercoiling. Consider using another description.
2. Page 3 line 111-112. “Latter interaction ...”. This sentence is confusing.
3. Page 5, line 183-184. “They also indicate...” Please clarify this statement.
4. Page 6, line 244. “ However, while MRN binding ...” Please reword this passage. It is confusing.
5. Page 7, line 256. “...binding near or at an DNA end” Change “an” to “a”

Reviewer #2

(Remarks to the Author)

The MRN complex plays important roles in DSB repair and signaling by sensing and resecting damaged DNA ends, and recruiting ATM. In the present work, Fan et al determined the structures of the MRN (or MR) complexed with DNA and proposed the “sensing” model. They showed how Nbs1 and ATM can compete for the binding to Rad50 “S” site and thereby switching between resection and signaling at the molecular level. They also determined the structures of the MRN (MR) bound to TRF2 (or TRF2 fragment) and DNA to provide a structural basis for inhibition of ATM signaling and DNA cleavage at telomeric DNA. These studies help to understand how human MRN complex recognizes DNA, and regulates the resection and signaling, and therefore beneficial to the DNA repair community. However, there are several places in the manuscript that are overinterpreted, which require clarification. In particular, the biological meaning of the MRN-TRF2-DNA complex structure should be carefully addressed.

Q1. Line 124–126 “.. our structural results argue for a topological sensing of linear DNA through ATP dependent loading coupled with ring-rod transition as observed for the bacterial MR...”

Authors claim that the hMRN senses DNA via coupling between the ring and rod transition. However, there is no evidence of such transition in the present work. All presented structures are clamp forms (could be “rod” shape in authors’ definition as shown in Fig 6) without the rest of CCD (majority portion). No other full length human MRN structures that describe “resting” or “cutting” state have been reported. The only full-length MRN structure with CCD is a DNA-free CtMRN, which can be considered as a rod-shape (Fig 3C). The structural differences described could be species-specific. Thus, I am not convinced that authors’ claims on the ring-rod transition. For the same reasons, authors proposal on DNA passage (single- and double-pass) proposed in line 127 to 130 are all hypothetical at present.

Q2. Line 145–148 “...The conformational differences suggest that DNA binding to RAD50 NBD leads to rearrangement of both lobes to properly align the ATPase motifs for catalysis as basis for DNA stimulated ATPase activity...”
The structural difference (Fig 3A vs 3C) observed here could be due to the species-specific (Ct vs. Hs Rad50) rather than DNA binding effects. The hMRN (hMR)-DNA structure should be compared to that of the hMRN (hMR) lacking DNA. Moreover, Ct Rad50-DNA in Fig 3b is a truncated structure lacking the large portion of CCD, thereby cannot restrain and clamp DNA. Therefore, this cannot be the proper comparison.

Q3. Line 157 – 159 & 266- 269 “...However, in contrast to bacterial MR, closing of the CCDs to a rod does not yet reposition the MRE11 nuclease from the auto-inhibited state to the cutting state ...”
To this reviewer, the contrast observed here is because authors have used ADP-BeFx which blocks the movement of the Mre11-Rad50 with respect to DNA whereas in the “cutting state” of the bacterial MR, ADP (Mg/Mn) has been used. Otherwise, it would be difficult to believe that the “cutting” state is only present and observed in bacterial MR.

Q4. It is reasonable for authors to define “sensing state” for the present structure. But the structure is basically similar to those from archaeal and yeast MR-DNA except they lack large portions of CCD clamping DNA. Which state would authors define the Ct MR-DNA complex (Fig 3b).

Q5. It would be helpful if the authors make marks the cleavage sites on the DNA bound to MRN? Also which parts of the substrate DNA are visible in the structures?

Q6. Line 192 to 194: authors conclude that Nbs1 binding to ATM (requires dissociation from Rad50) is an allosteric transition from DNA sensing to signaling state. Is there any biological evidence to support the allosteric transition?

Q7. Line 214 (line 217 - 225). Only density for 14 residues for iDDR was observed. Yet when iDDR motif was added to the MRN complex for structural analysis, it did not make a complex. Are there any significant differences in the affinities between the peptides (or TRF2) and MR (MRN)?

Q8. The present structure and biochemical data show that TRF2 tightly binds to MRN. How does such tight binding (or TRF2 binding) prevents the cleavage of telomere by the MRN complex.

Q9. Previous study (by Khayat et al. 2024) reports that iDDR binding to Rad50 disrupts the binding of Rad50 to DNA, which is not the case in this study. Authors may comment on the differences.

Q10. Line 235- 237; Authors observed that in the presence of TRF2, Mre11 is detached from Rad50, which is a putative path for the “cutting” state based on the authors claim. However, TRF2’s main role is to protect the telomere from the cleavage by MRN, which is rather opposite to the present observation.

Q11. The Nbs1 C-helix, iDDR and ATM competitively bind to the “S-site” of Rad50. What controls the binding of these components? It would be helpful if authors discuss on this issue.

Q12. Fig 6. A model in Fig 6 is derived from the all reported bacterial/eukaryotic MR and present study including the partial- and full-length structures. While this reviewer appreciates authors’ efforts to propose a model for the MRN functions, the proposed model could mislead the readers. For example, for the resting state, there is no evidence of the open (ring) form of CCD is connected to the ring form of the head (although this figure has been used for long time in the community). The rod and ring form of the zinc-hook and head could be combined and mixed in all proposed models. Even in the sensing state in Fig 6, the only precise part is a head region. In this reviewer’s opinion, authors should clearly state that some models are derived from bacterial or different species, and should tone down the model.

Q13. Authors also proposed a deprotected form for their MRN-TRF2-DNA structure. However, there are other shelterin component which binds at the same site. Thus, it is unclear to me the biological meaning of the deprotected form by the TRF2.

Q14. Line 281/282 – I could not find the paper. Please check it.

Reviewer #3

(Remarks to the Author)

The authors report the cryo-electron microscopy structures of human MR-DNA, MRN-DNA, MR-TRF2iDDR-Myb-DNA,

MRN-TRF2iDDR-Myb-DNA, and MRN-TRF2-DNA. Among all these structures, the active sites of MRE11 nuclease dimer are blocked by RAD50, decoupling of DNA binding and nuclease activity, which is different from bacterial homolog structures. DNA-binding would trigger the rearrangement of nucleotide binding site of RAD50 and neither NBS1 nor TRF2 binding will change the structure of MR-DNA part.

There are multiple new structural observations. The C-terminal helix (718-745) of NBS1 is found to interact with both MRE11 and the S site of RAD50, stabilizing the interaction between MRE11 and RAD50 and blocking the recruitment for ATM. The iDDR motif of TRF2 binds with the S site of RAD50 to inhibit MRE11's nuclease activity, but it could only occupy the second S site when NBS1 is present. Structural results and insights are developed for topological DNA end sensing, DNA tethering, and intrinsic vs extrinsic MRE11 nuclease regulation.

Overall, the results and insights reported are tours de force that will change how MRN function is understood and how future experiments will be designed. The authors report important findings of NBS1 C-terminal region and iDDR motif of TRF2 regulating MRN activities. They propose the new model for better understanding the mechanism of human MRN function. Coupled consideration of the signature motif surrounding the RAD50 nucleotide-binding site and the topologically detection of DSBs by ATP-dependent transitions of the coiled-coil domain nicely explains the unusual RAD50 split ATPase domain with its inserted coiled-coil structure. The manuscript is written well, the methods are appropriate, and the work is generally presented in a manner that should support reproducibility.

The authors should consider the following points to improve the manuscript:

1. In page#5 line#167, Fig.S9C was mentioned to support "NBS1 does not induce significant changes into the DNA bound MR structure, compared to MR alone", but no structural superimposition of MR-DNA and MRN-DNA was found, which would help readers.
2. In Figure 5E, TRF2 slightly decreased the ATPase activity of MR while TRF2iDDR-Myb doesn't change the ATPase activity of MRN, which doesn't support page6 line#235 very well. In this experiment, two different TRF2 constructs were used, which may cause the difference. The same construct should be used in the experiment. In addition, the time to measure the ATPase activity of MR/MRN + TRF2 (90min) is longer than MR/MRN (60min) in page#12 line#475. Are there any special reasons to do so? Perhaps this could be clarified.
3. For Figure S9A and other MRN fluorescence anisotropy assay, the highest concentration of MRN used is 300nM. Is this protein concentration is enough to reach the plateau. TRF2 interacts with RAD50 and has very high affinity for telomeric DNA based on Figure 5F. Combining with Figure S9A, there is a question as to whether pre-loading DNA onto TRF2 may help MRN assembling on telomeric DNA.

Minor:

1. The structural comparison of MR-TRF2iDDR-Myb-DNA and MRN-TRF2iDDR-Myb-DNA gives the idea that NBS1 could outcompete TRF2iDDR-Myb for the S site of RAD50. A competitive binding assay would be better added to support this.
2. In Figure 1B and Figure 5E, NBS1 decreased the ATPase activity of RAD50 compared to MR only, but in the manuscript, this was not mentioned. The authors should address this difference.
3. MRE11 mutant was used in the cryo-EM sample. It's better to include in the Table1.
4. Some incorrect numbers need to be corrected. E.g., page#5 line# 171, "L732NBS1" should be L723NBS1; page#13 line#524, "573,534 particles" is different from Figure S2; page#14 line#594, "3.01 Å" is different from Figure S8.
5. Some typos need to be corrected. E.g., page#9 line#359, "MH129RN" should be "MH129NRN"; line#366 "mio/mL" should be corrected; line#373, "16,000 rom" should be 16,000 rpm; Supplementary figures and information page#21 line#373, "(2.97Q Å)" should be 2.97 Å.
6. Figure S9A and S9B were not labelled.
7. Where is the S site defined - does S stand for surface?

Reviewer #4

(Remarks to the Author)

Version 1:

Reviewer comments:

Reviewer #2

(Remarks to the Author)

Authors have clearly resolved all of this reviewer's concerns in the revised manuscript.

Reviewer #3

(Remarks to the Author)

The authors further strengthened their manuscript with new data, clear insights, and revised text. They have appropriately addressed all reviewer concerns. The structures appear definitive.

In my view they did an exemplary job in both their responses to reviewer points and their revisions to the text.

The revised manuscript is without doubt an outstanding research advance that as the title states defines DNA Break Sensing by Human MRE11-RAD50-NBS1 and its Regulation by Telomeric Factor TRF2. It will be of general and long lasting interest.

Reviewer #4

(Remarks to the Author)

We thank all reviewers for their careful evaluation of our manuscript, their positive feedback and constructive and insightful criticism, which have significantly improved our work. We now have addressed all points and revised the manuscript accordingly. The most important revisions are:

- New structural data on the “resting state” of human MRN with open CCDs, obtained in the presence of ATPγS. These provide important experimental evidence for ring–rod transitions and support our model of topological DNA sensing in human MRN. These data are also added to EMDB.
- Revised ATPase activity measurements using harmonized MR/MRN and TRF2 constructs.
- Updated binding experiments with TRF2 preloaded on DNA and MRN titration.
- Revised and expanded results and discussion along the reviewers suggestions.

Details are provided in the point-by-point response below. Changes to the manuscript are color highlighted, except for some trivial spelling mistakes. We thank all reviewers again for their time and input. We hope that our revisions satisfactorily address the reviewers’ points, and the manuscript is ready for publication.

Reviewer #1 (Remarks to the Author)

Fan et al determine cryo-EM structures of the human MR and MRN complexes with DNA and the Shelterin component TRF2. This stunning work reveals the complex molecular logic behind the regulation of MRN nuclease activity at telomeres. Overall, the work provides a clear molecular explanation for decades of work from multiple labs on the regulation of human MRN activities, and how this layer of complexity differs from bacterial MR orthologs. The paper is clearly written and illustrated, and near ready for publication already. I have minor points.

Points:

1. The use of the “DNA topology” and “topological sensing” throughout the text is confusing in the context of how we normally define DNA topology in terms of supercoiling. Consider using another description.

We thank the reviewer for the enthusiastic feedback! We of course agree that DNA topology is historically rooted in supercoiling. However, “topological entrapping” and “topological loading” have become key terms in the case of SMC proteins, and they are often used to represent the same idea, namely that a DNA duplex or loop is threaded through a proteinaceous ring structure (please see e.g. DOI: 10.1002/bies.202400120). For that reason, we believe that “topology” is principally a good term to also highlight similarities between Rad50 and SMC proteins. To address this comment, we a) reduced the usage of “topology”, b) explained it more appropriately to avoid confusion, and c) use “gating” as an alternative when appropriate. We hope these changes appropriately address the reviewer’s point.

2. Page 3 line 111-112. “Latter interaction ...”.

This sentence is confusing.

We deleted this sentence (a minor point) to improve clarity and conciseness

3. Page 5, line 183-184. “They also indicate...”

Please clarify this statement.

We deleted this sentence (a speculative minor point) to improve clarity and conciseness

4. Page 6, line 244. "However, while MRN binding ..."
Please reword this passage. It is confusing.

Thanks for pointing this out. Changed to: "We could not reach saturating binding conditions in our assays of MRN due to a general limit to concentrate MRN. Possible causes are multiple binding events or clustering of MRN, as has been observed for MRX (doi.org/10.1038/s41467-022-29841-0). However, we obtain a properly saturated binding curve in the presence of TRF2 (Fig. S4F) suggesting that TRF2 promotes a more defined binding of MRN."

5. Page 7, line 256. "...binding near or at an DNA end"
Change "an" to "a"

Corrected, thanks

Reviewer #2 (Remarks to the Author)

The MRN complex plays important roles in DSB repair and signaling by sensing and resecting damaged DNA ends, and recruiting ATM. In the present work, Fan et al determined the structures of the MRN (or MR) complexed with DNA and proposed the "sensing" model. They showed how Nbs1 and ATM can compete for the binding to Rad50 "S" site and thereby switching between resection and signaling at the molecular level. They also determined the structures of the MRN (MR) bound to TRF2 (or TRF2 fragment) and DNA to provide a structural basis for inhibition of ATM signaling and DNA cleavage at telomeric DNA. These studies help to understand how human MRN complex recognizes DNA, and regulates the resection and signaling, and therefore beneficial to the DNA repair community. However, there are several places in the manuscript that are overinterpreted, which require clarification. In particular, the biological meaning of the MRN-TRF2-DNA complex structure should be carefully addressed.

Q1. Line 124–126 "...our structural results argue for a topological sensing of linear DNA through ATP dependent loading coupled with ring-rod transition as observed for the bacterial MR..."
Authors claim that the hMRN senses DNA via coupling between the ring and rod transition. However, there is no evidence of such transition in the present work. All presented structures are clamp forms (could be "rod" shape in authors' definition as shown in Fig 6) without the rest of CCD (majority portion). No other full length human MRN structures that describe "resting" or "cutting" state have been reported. The only full-length MRN structure with CCD is a DNA-free CtMRN, which can be considered as a rod-shape (Fig 3C). The structural differences described could be species-specific. Thus, I am not convinced that authors' claims on the ring-rod transition. For the same reasons, authors proposal on DNA passage (single- and double-pass) proposed in line 127 to 130 are all hypothetical at present.

The reviewer's point prompted us to experimentally investigate the "resting" state of hMRN. In the original submission, we did provide some evidence for the "resting" state with open CCDs, but only as 2D class averages (Fig. S5A, S5B large panels). However, we collected more data on hMRN in the presence of ATP γ S. This allowed us to calculate a 3D reconstruction that clearly shows open CCDs, altogether looking very similar to the "resting state" observed for SbcCD (<https://doi.org/10.1016/j.molcel.2019.07.035>). Due to orientational bias and flexibility between Rad50 and Mre11 in this resting state, the map remains anisotropic, preventing detailed model building at this stage. However, the overall map almost perfectly matches an AlphaFold3 prediction of hMRN (with shortened CCDs), including the correctly predicted coiled-coil angles. The map also

shows density for ATP γ S. This analysis defines important features of the resting state of hMRN, such as a retained interaction of the C-helix with the S-site of Rad50. All in all, in our opinion these data provide experimental evidence for a resting state in human MRN that is very similar to that of SbcCD.

Furthermore, open and closed CCDs have been observed indeed before with full length hMRN. They have been visualized by atomic force microscopy (AFM). Away from DNA, hMRN exhibited predominantly open CCDs, but when bound to DNA, they were closed along their entire length. The AFM images are also consistent with a single passage of DNA through the CCDs when closed (Please see doi: 10.1038/nature03927, in particular Fig. 4.)

Altogether, we thank the reviewer for raising this point since it prompted us to investigate and improve our analysis of the resting state. We hope these improved analyses along with available literature data provide sufficient evidence now for the ring and rod states as proposed in our model. We added these new data to Fig. 2B, Fig. S5, Fig. S11 and toned our manuscript in the discussion.

Q2. Line 145–148 “...The conformational differences suggest that DNA binding to RAD50 NBD leads to rearrangement of both lobes to properly align the ATPase motifs for catalysis as basis for DNA stimulated ATPase activity...”

The structural difference (Fig 3A vs 3C) observed here could be due to the species-specific (Ct vs. Hs Rad50) rather than DNA binding effects. The hMRN (hMR)-DNA structure should be compared to that of the hMRN (hMR) lacking DNA.

Moreover, Ct Rad50-DNA in Fig 3b is a truncated structure lacking the large portion of CCD, thereby cannot restrain and clamp DNA. Therefore, this cannot be the proper comparison.

We appreciate the reviewer’s point and rewrote this part more carefully. Our new low-resolution structure of hMRN in the presence of ATP γ S but absence of DNA indeed reveals some species-specific differences:

- i) We found that ATP γ S is not sufficient to open the CCDs in the *Chaetomium* MRX complex (doi: 10.1016/j.molcel.2022.12.003) but do see some open CCDs for *Chaetomium* MRX in the presence of ATP (Fig. S4b in doi: 10.1016/j.molcel.2022.12.003)
- ii) We do not see formation of “sensing” state clamps with encircled DNA in the presence of ATP γ S in the case of hMRN, but it is observed in the case of *Saccharomyces* MR (Preprint: doi: 10.21203/rs.3.rs-5390974/v1.).

We believe these differences are unlikely due to fundamental differences in the mechanism of loading and sensing between MRN complexes of different species. In our opinion, they more likely represent species adaptations resulting in differing interface strengths, in particular the interaction strength between the two CCDs. Such changes and/or subtle differences in the ATP binding site - which for instance might reflect the different optimal temperatures for yeast, human and *Chaetomium thermophilum* - could determine how well ATP analogs are functioning the same way as ATP in opening the CCDs and enabling sensing state clamp formation.

Nevertheless, the reviewer is correct in pointing out that comparing DNA bound hMRN to apo CtMRX is not a valid comparison at this point. We therefore

- a) removed the part on DNA stimulated ATPase.
- b) removed CtRad50-DNA-ATP γ S (PDB: 5DAC) from figure 3b.
- c) included the experimental hMRN resting state in figure 2b.
- d) added a part in the discussion that deals with these species-specific differences

Q3. Line 157 – 159 & 266- 269 “...However, in contrast to bacterial MR, closing of the CCDs to a rod does not yet reposition the MRE11 nuclease from the auto-inhibited state to the cutting state ...” To this reviewer, the contrast observed here is because authors have used ADP-BeFx which blocks the movement of the Mre11-Rad50 with respect to DNA whereas in the “cutting state” of the bacterial MR, ADP (Mg/Mn) has been used. Otherwise, it would be difficult to believe that the “cutting” state is only present and observed in bacterial MR.

We appreciated the reviewer’s comment which requires us to reformulate relevant parts more clearly.

- 1) Please note that in contrast to the auto-activated bacterial SbcCD, formation of stable cutting states in eukaryotic MRN/X likely requires an additional protein (CtIP in humans, Sae2 in yeast), which is absent in prokaryotes. This has been recently investigated through a combination of structure prediction, mutagenesis and biochemistry (doi: 10.1016/j.molcel.2024.05.019).
- 2) Please also note that for bacterial SbcCD, we did not use ADP but ATP to assemble the cutting state. Using ADP from the beginning does not result in DNA binding at all. SbcCD critically needs ATP and which is then hydrolysed to ADP after DNA loading. To test a similar reaction in hMRN, we omitted BeFx and used only ATP during incubation. The resulting structure practically looks like the ATP+BeFx structure, a sensing state with closed CCDs. Again, and unlike SbcCD, this only leads to the clamping state and not a cutting state. Although the map is anisotropic, it is clear in the active site indicating that ATP has been hydrolysed to ADP+Pi. It is plausible that formation of a cutting state would need phosphate release in conjunction with CtIP, however this is currently beyond our reach and needs to be addressed in future studies.

Thus, we currently believe that ATP+BeFx, resulting in ADP-BeFx mimics somehow this posthydrolysis state while ATP γ S mimics more ATP (open CCDs). We adjusted the discussion to compare bacterial and human complexes more clearly, point out the role of CtIP, like in Q2-d. The need to better clarify the role of ATP hydrolysis was already part of the limitations, and we need to address the underlying mechanism in the future. The provided figure of the preliminary structural studies is for the consideration by the reviewer.

Q4. It is reasonable for authors to define “sensing state” for the present structure. But the structure is basically similar to those from archaeal and yeast MR-DNA except they lack large portions of CCD clamping DNA. Which state would authors define the Ct MR-DNA complex (Fig 3b).

- 1) Please note the archaeal structures would represent the resting state, or a loading state where DNA binds to the MR head with open CCDs. They do not represent the sensing state with close CCDs.
- 2) The ‘sensing state’ of the presented structures is defined with (1) closed RAD50 CCDs upon DNA binding; (2) yet not repositioned MRE11 nuclease. Therefore, the CtMRN structure is more appropriately to be defined as another “resting state” with the notable difference that the CCDs are closed compared to the hMRNs “resting state”. In our structural determination of CtMRN, we observed also open CCDs in the presence of ATP, like the hMRN resting state (please see doi: 10.1016/j.molcel.2022.12.003, Fig. S4D). At this point we have only the explanation presented above: We believe these differences are unlikely due to fundamental differences in the mechanism of loading and sensing between MRN complexes of different species. In our opinion, they more likely represent differences in the interaction strength between the two CCDs. Such changes might impact how well ATP or ATP analogs open the CCDs in human and *Chaetomium thermophilum* MRN.

We added latter point to the discussion (see also Q2, Q3).

Q5. It would be helpful if the authors make marks the cleavage sites on the DNA bound to MRN? Also which parts of the substrate DNA are visible in the structures?

We are afraid but we can't provide an answer to this question. Since we do not know the cutting state, we do not know the precise cleavage sites. We could estimate them based on the SbcCD cutting states, but this would be meaningless, since we do not visualize the termini of the DNA. We built as much DNA as we can see in well-defined density, but DNA proceeds on both sides beyond that (increasingly smeared, fuzzy density. All we can say that MRN binds to DNA somewhat away from the termini on our 50bp substrate. To properly answer this point, we indeed need to wait until we (or others) have managed to get a cutting-state-like structure.

Q6. Line 192 to 194: authors conclude that Nbs1 binding to ATM (requires dissociation from Rad50) is an allosteric transition from DNA sensing to signaling state. Is there any biological evidence to support the allosteric transition?

That is a good question! It is clear that Nbs1's C-helix needs to dissociate from Rad50 in order to bind ATM, but we currently do not have direct experimental evidence for conformational changes in MRN. We referred to this C-helix removal as an allosteric transition, but perhaps this is misleading. We now reformulated to “C-helix switch”.

Q7. Line 214 (line 217 - 225). Only density for 14 residues for iDDR was observed. Yet when iDDR motif was added to the MRN complex for structural analysis, it did not make a complex. Are there any significant differences in the affinities between the peptides (or TRF2) and MR (MRN)?

We currently have no means to provide quantitative measurements of the interaction. In fact, we do not see an interaction in gel filtration indicating $>\mu\text{M}$ affinities of iDDR to the S-site. Such low affinities are very difficult to measure since they typically require protein concentrations in that

range. This is not possible with MR/MRN due to a high stickiness to surfaces and aggregation tendency, which currently prevents methods such as ITC, BLI or SPR (we tried).

The observed differences can be, however, easily explained through avidity (i.e. accumulated strength): if iDDR is coupled to the Myb domain through a short linker, as naturally in TRF2, both Myb and MRN would bind with high affinity to the same DNA. Essentially it will increase the local concentration of iDDR in the vicinity of MRN when both are bound adjacent on telomeric DNA and thus ensure population of the interaction despite low Kds. We explained this point better in the Results part and hope this properly addresses the reviewers' point.

Q8. The present structure and biochemical data show that TRF2 tightly binds to MRN. How does such tight binding (or TRF2 binding) prevent the cleavage of telomere by the MRN complex.

TRF2 simply blocks the interaction with CtIP, which will prevent formation of the cutting state at telomeric DNA. Several publications on TRF2 or an equivalent protein in yeast Rif2 proposed this mechanism independently (Khayat et al. 2024; Myler et al. 2023; Marsella et al. 2021). We provide an experimental structure and show that this mechanism can occur in the sensing state, suggesting that MRN is prevented from adopting cutting states by steric hindrance of CtIP interaction.

Q9. Previous study (by Khayat et al. 2024) reports that iDDR binding to Rad50 disrupts the binding of Rad50 to DNA, which is not the case in this study. Authors may comment on the differences.

In Khayat et al., the authors did not use the full MRN complex for their studies, only the Rad50^{NBD}. This will result in different binding to DNA and much faster k_{off} rates, since there are no CCDs hence no clamping interactions. Thus, it is possible that Khayat et al. probed more subtle changes of iDDR to Rad50 that may not be as relevant in full MRN. Alternatively, biochemical evidence by others that MRX clusters at DSBs, and Rad50-Rad50 interactions play an important role in this clustering (doi: 10.1038/s41467-022-29841-0). In Khayat et al., iDDR could reduce Rad50-Rad50 interactions and therefore lead to less binding. We also see differences in binding curves of MRN to DNA in the presence or absence of TRF2, so there appears to be an effect in our case as well. A plausible explanation would be a reduction of clustering and a more defined binding due to the iDDR interaction of MRN. In any case, we arguably probe a physiologically much more relevant scenario since we use full length MRN and not only Rad50^{NBD}. We expanded on this important point in the discussion and thank the reviewer for the observation.

Q10. Line 235- 237; Authors observed that in the presence of TRF2, Mre11 is detached from Rad50, which is a putative path for the "cutting" state based on the authors claim. However, TRF2's main role is to protect the telomere from the cleavage by MRN, which is rather opposite to the present observation.

This aspect was perhaps not clear enough. We observe the partial detachment regardless of whether TRF2 and/or Nbs1 is present. Thus, TRF2 neither prevents nor stimulates it. We altered this part in the manuscript response to reviewer 3 as well, hope this resolves this point.

Q11. The Nbs1 C-helix, iDDR and ATM competitively bind to the "S-site" of Rad50. What controls the binding of these components? It would be helpful if authors discuss on this issue.

We thank the reviewer for this point but we do not know at this point if and what is the regulation. There are no obvious candidates such as phosphorylation-sites (like in the case of CtIP). Even in our new ATP γ S structure with open CCDs, the NBS1 C-helix still interacts with the S-site, suggesting the dissociation of NBS1 C-helix is not a simple process controlled by ring and rod states. In the case of

TRF2, we do not believe regulation is needed since TRF2 simply blocks the free S-site and prevents CtIP binding to promote the cutting state. The situation is different in the case of ATM. A possible model is that ATP binding and hydrolysis cycles induce some sort of dynamics (which we do not see in our static cryo-EM studies at this point) in the C-helix that can lead to a “capture” of transiently displaced C-helix by ATM. We expanded on this point in the discussion.

Q12. Fig 6. A model in Fig 6 is derived from the all reported bacterial/eukaryotic MR and present study including the partial- and full-length structures. While this reviewer appreciates authors’ efforts to propose a model for the MRN functions, the proposed model could mislead the readers. For example, for the resting state, there is no evidence of the open (ring) form of CCD is connected to the ring form of the head (although this figure has been used for long time in the community). The rod and ring form of the zinc-hook and head could be combined and mixed in all proposed models. Even in the sensing state in Fig 6, the only precise part is a head region. In this reviewer’s opinion, authors should clearly state that some models are derived from bacterial or different species, and should tone down the model.

We would like to point the reviewer to e.g. doi: 10.1038/nature03927, where clear evidence of ring and rod forms are displayed using atomic force microscopy. We do not want to rule out some intermediates, e.g. part of the CCD is in rod, part is in open states, but contrary to the folded rod and open ring, this has not been observed yet. We also add now structural evidence for the open CCDs to our manuscript, which strengthens the resting state depiction. We now add evidence for the resting state with open CCDs (Fig. 2D, S10), which is very similar to bacterial MR resting state. To address the reviewer’s point, we more clearly stated the literature evidence, and wrote that parts of the model such as a proposed cutting state are compiled from AFM studies along with AlphaFold predictions as suggested.

Q13. Authors also proposed a deprotected form for their MRN-TRF2-DNA structure. However, there are other shelterin component which binds at the same site. Thus, it is unclear to me the biological meaning of the deprotected form by the TRF2.

This is perhaps a misunderstanding. With “deprotected” we did not mean telomeres with no bound protein, but dysfunctional telomeres, e.g. without a T-loop and perhaps with some exposed dsDNA, where MRN can bind onto. In the literature, the term “dysfunctional” is often used as well, so we changed to that term.

Q14. Line 281/282 – I could not find the paper. Please check it.

Corrected, thanks.

Reviewer #3 (Remarks to the Author):

The authors report the cryo-electron microscopy structures of human MR-DNA, MRN-DNA, MR-TRF2iDDR-Myb-DNA, MRN-TRF2iDDR-Myb-DNA, and MRN-TRF2-DNA. Among all these structures, the active sites of MRE11 nuclease dimer are blocked by RAD50, decoupling of DNA binding and nuclease activity, which is different from bacterial homolog structures. DNA-binding would trigger the rearrangement of nucleotide binding site of RAD50 and neither NBS1 nor TRF2 binding will change the structure of MR-DNA part.

There are multiple new structural observations. The C-terminal helix (718-745) of NBS1 is found to interact with both MRE11 and the S site of RAD50, stabilizing the interaction between MRE11 and RAD50 and blocking the recruitment for ATM. The iDDR motif of TRF2 binds with the S site of RAD50

to inhibit MRE11's nuclease activity, but it could only occupy the second S site when NBS1 is present. Structural results and insights are developed for topological DNA end sensing, DNA tethering, and intrinsic vs extrinsic MRE11 nuclease regulation.

Overall, the results and insights reported are tours de force that will change how MRN function is understood and how future experiments will be designed. The authors report important findings of NBS1 C-terminal region and iDDR motif of TRF2 regulating MRN activities. They propose the new model for better understanding the mechanism of human MRN function. Coupled consideration of the signature motif surrounding the RAD50 nucleotide-binding site and the topological detection of DSBs by ATP-dependent transitions of the coiled-coil domain nicely explains the unusual RAD50 split ATPase domain with its inserted coiled-coil structure. The manuscript is written well, the methods are appropriate, and the work is generally presented in a manner that should support reproducibility.

The authors should consider the following points to improve the manuscript:

1. In page#5 line#167, Fig.S9C was mentioned to support "NBS1 does not induce significant changes into the DNA bound MR structure, compared to MR alone", but no structural superimposition of MR-DNA and MRN-DNA was found, which would help readers.

We added the requested superposition to Supplementary Fig. 11C.

2. In Figure 5E, TRF2 slightly decreased the ATPase activity of MR while TRF2iDDR-Myb doesn't change the ATPase activity of MRN, which doesn't support page6 line#235 very well. In this experiment, two different TRF2 constructs were used, which may cause the difference. The same construct should be used in the experiment. In addition, the time to measure the ATPase activity of MR/MRN + TRF2 (90min) is longer than MR/MRN (60min) in page#12 line#475. Are there any special reasons to do so? Perhaps this could be clarified.

We thank the reviewer which prompted us to repeat the ATPase experiments with harmonized components. We used the TRF2^{iDDR-Myb} construct for both, MRN and MR. The outcome shows that ATP hydrolysis rates are not different between MRN, MR and TRF2 complexes. The new data replace the original ones in our revised manuscript. Also, we want to point out that we used now consistent 64 bp Myb DNA as the substrate for comparability in all measurements, whereas we used 50 bp DNA for MR(N) DNA stimulated ATPase assays in our original draft.

3. For Figure S9A and other MRN fluorescence anisotropy assay, the highest concentration of MRN used is 300nM. Is this protein concentration is enough to reach the plateau. TRF2 interacts with RAD50 and has very high affinity for telomeric DNA based on Figure 5F. Combining with Figure S9A, there is a question as to whether pre-loading DNA onto TRF2 may help MRN assembling on telomeric DNA.

In fact, we are limited by the available amounts of MRN and to how much the protein could be concentrated. Since both MR and MRN are highly prone to aggregation, we had to use fresh proteins (without freeze and thaw cycles) for experiments. The fluorescence anisotropy assay with MRN and TRF2 was indeed carried out by preloading TRF2 onto DNA for 15 minutes at room temperature, then adding MRN. We may, in fact, see clustering of MRN alone on DNA, which could account for the lack of saturation but also increased anisotropy compared to the presence of TRF2. We performed revised binding experiments which show similar effects and discussed this point more carefully. Figures 2C, 4F and S11 are updated.

Minor:

1. The structural comparison of MR-TRF2iDDR-Myb-DNA and MRN-TRF2iDDR-Myb-DNA gives the idea that NBS1 could outcompete TRF2iDDR-Myb for the S site of RAD50. A competitive binding assay would be better added to support this.

In a sense, our cryo EM experiments are a qualitative competitive binding assay since we observe TRF2^{iDDR-Myb} only on the Rad50 S site not occupied by NBS1 in the case of MRN, but bound to both RAD50 S sites in the case of MR. These findings establish that the presence of NBS1 does not permit binding of TRF2^{iDDR-Myb} under conditions where iDDR can readily bind a free S site. As MRN has limited concentrations at physiological buffers, prone to aggregation, stickiness to surface, we have not been able to set up binding assays of MRN using SPR nor Microcalorimetry (we tried). To address the reviewer's point, we still adjusted our wording and wrote that NBS1 blocks the binding of TRF2 at one S-site and avoided the term "competition".

2. In Figure 1B and Figure 5E, NBS1 decreased the ATPase activity of RAD50 compared to MR only, but in the manuscript, this was not mentioned. The authors should address this difference.

We thank the reviewer which prompted us to repeat the ATPase experiments with harmonized components. The outcome shows that ATP hydrolysis rates are not different between MRN, MR and TRF2 complexes. The new data replaces the original ones in our revised manuscript.

3. MRE11 mutant was used in the cryo-EM sample. It's better to include in the Table1.

Included, thanks.

4. Some incorrect numbers need to be corrected. E.g., page#5 line# 171, "L732NBS1" should be L723NBS1; page#13 line#524, "573,534 particles" is different from Figure S2; page#14 line#594, "3.01 Å" is different from Figure S8.

Corrected, thanks.

5. Some typos need to be corrected. E.g., page#9 line#359, "MH129RN" should be "MH129NRN"; line#366 "mio/mL" should be corrected; line#373, "16,000 rom" should be 16,000 rpm; Supplementary figures and information page#21 line#373, "(2.97Q Å)" should be 2.97 Å.

Corrected, thanks.

6. Figure S9A and S9B were not labelled.

Corrected, thanks.

7. Where is the S site defined - does S stand for surface?

We were inspired by Kayat et al. who also use "S" region. The "S" comes from Rad50S (separation of function) mutations that cluster on a Rad50 surface patch. In essence, these residues, discovered through a screen in yeast, interfere with regulator binding (Sae2/CtlP, Rif2/TRF2). We found it more elegant to refer to as "S site" instead of Rad50S surface cluster.

We thank the editor and all reviewers for their positive feedback on the revised manuscript. We hope that we address all remaining issues in the final version of the manuscript.

Point-by-point response to referees:

Reviewer #2 (Remarks to the Author):

Authors have clearly resolved all of this reviewer's concerns in the revised manuscript.

We thank the reviewer for the kind appreciation of our manuscript.

Reviewer #3 (Remarks to the Author):

The authors further strengthened their manuscript with new data, clear insights, and revised text. They have appropriately addressed all reviewer concerns. The structures appear definitive.

In my view they did an exemplary job in both their responses to reviewer points and their revisions to the text.

The revised manuscript is without doubt an outstanding research advance that as the title states defines DNA Break Sensing by Human MRE11-RAD50-NBS1 and its Regulation by Telomeric Factor TRF2. It will be of general and long lasting interest.

We thank the reviewer for the kind appreciation and recommendation of our revised manuscript.